# Incentivizing Quality Text Generation via Statistical Contracts

**Eden Saig**[1], **Ohad Einav**[1], **Inbal Talgam-Cohen**[1,2]
[1] Technion – Israel Institute of Technology
[2] Tel Aviv University
{edens,ohadeinav,italgam}@cs.technion.ac.il

## Abstract

While the success of large language models (LLMs) increases demand for machine-generated text, current pay-per-token pricing schemes create a misalignment of incentives known in economics as *moral hazard*: Text-generating agents have strong incentive to cut costs by preferring a cheaper model over the cutting-edge one, and this can be done "behind the scenes" since the agent performs inference internally. In this work, we approach this issue from an economic perspective, by proposing a pay-for-performance, contract-based framework for incentivizing quality. We study a principal-agent game where the agent generates text using costly inference, and the contract determines the principal's payment for the text according to an automated quality evaluation. Since standard contract theory is inapplicable when internal inference costs are unknown, we introduce *cost-robust* contracts. As our main theoretical contribution, we characterize optimal cost-robust contracts through a direct correspondence to optimal composite hypothesis tests from statistics, generalizing a result of Saig et al. (NeurIPS'23). We evaluate our framework empirically by deriving contracts for a range of objectives and LLM evaluation benchmarks, and find that cost-robust contracts sacrifice only a marginal increase in objective value compared to their cost-aware counterparts.

## 1 Introduction

Modern-day LLMs are showing increasingly impressive capabilities, and simultaneously becoming increasingly costly. With rising success at handling complex tasks, conversational AI systems are seeing ubiquitous usage across critical domains such as healthcare [19, 34], financial risk assessment [27], and law [24, 35]. To achieve such levels of performance, contemporary LLM architectures contain billions and even trillions of parameters, leading to a computational pipeline that requires dedicated facilities and substantial energy to operate [33].

Due to the high computational requirements of modern LLMs, language generation tasks are typically outsourced to commercial firms which generate text for a fee. These firms either maintain dedicated infrastructure optimized for inference workloads, or act as intermediaries that facilitate access to such resources. To address the tension between performance and computational costs, such firms typically have multiple service options, each offering a different trade-off between model quality and cost [1, 7, 36, 37]. Currently, the most common pricing scheme for such services is *pay-per-token*, in which users agree in advance to pay a fixed rate for each token of text generated by the system [10].

While simple and intuitive, the pay-per-token pricing scheme creates a misalignment of economic incentives between the firms and their consumers, known in the economic literature as *moral hazard*: As inference is performed internally and a fixed price is agreed upon in advance, firms can strategically increase their profit margin by generating text using a cheaper, lower-quality model. Due to the stochastic nature of language generation, consumers may not be able to reliably determine the quality of the model being used, exposing them to this kind of hazard.

Moral hazard is especially prevalent in cases where the text generation task is complex, and so evaluation is hard: Consider a scenario where a healthcare provider hires a firm to use conversational AI for summarizing medical notes. As medical diagnosis is an intricate and critical task, the healthcare provider wishes the medical summaries to be generated using the most advanced language model. Under the pay-per-token pricing scheme, the healthcare provider agrees in advance to pay the firm a fixed amount for each token generated. However, it is not hard to imagine that the firm may attempt to increase profit margins by routing some of the summarization requests to cheaper language models, instead of the most advanced one, without taking into account their purpose, and knowing that any lower-quality results would be attributed to the stochastic nature of LLM inference.

**From pay-per-token to pay-for-performance.** In the economic literature, the canonical solution to moral hazard problems is *pay-for-performance*, or *P4P* [17]. Instead of paying a fixed price for any outcome, the parties agree in advance on a *contract* that specifies a differential payment scheme – for example, agreeing in advance that the firm will receive higher pay when the generated text is considered to be of higher quality. When designed correctly, contracts incentivize rational agents to invest more effort, thus providing a way to align incentives. Interaction around contracts is modeled as a principal-agent game, where the principal commits to a payment scheme, and the agent responds by rationally selecting a utility-maximizing action. Within this framework, the principal seeks to design a contract which satisfies some notion of optimality, such as requiring the least amount of expected pay ("min-pay contract"), or the lowest budget ("min-budget contract").

In this work, we extend the theory of contract design, and use it to design optimal pay-for-performance pricing schemes for delegated text generation. Applying contract design to this setting requires us to overcome the challenges of *automated evaluation* and *cost uncertainty*. The former stems from the need for a scalable measure of performance to support pay-for-performance pricing, while the latter arises from the principal's uncertainty about the agent's true internal cost structure, as commercial firms often regard operational costs and implementation details as proprietary information.

**Our results.** To tackle automated evaluation, we draw upon recent advances in the LLM evaluation literature [9], and propose a modular contract design framework which uses LLM evaluators as subroutines. More specifically, upon receiving generated text, our pricing scheme is implemented by evaluating the prompt-response pair using an automated evaluator and paying accordingly. The choice of evaluator can be tailored to the task: optimal pricing schemes in code generation tasks, for example, would rely on a pass/fail code evaluator [11, 4], whereas evaluation of linguistic tasks can be achieved using an "LLM-as-a-judge" approach [41, 28, 25]. In our theoretical analysis, we show that our framework is applicable even to intricate tasks where current evaluation methods are noisy and undecisive, as the principal can compensate for the noise by paying more (Proposition 1).

To address the challenge of cost uncertainty, we propose a new notion of *cost-robust contracts*, which are pay-for-performance schemes guaranteed to incentivize effort even when the internal cost structure is uncertain. Our main theoretical contribution is a statistical characterization of optimal cost-robust contracts (Theorem 1): We prove a direct correspondence between optimal cost-robust contracts and statistical hypothesis tests by showing that the min-budget and min-pay contract objectives correspond to minimax risk functions of composite hypothesis tests (Type-1+Type-2 errors and FP/TP, respectively). This significantly generalizes a recent result by Saig et al. [31] to arbitrary action spaces and multiple optimality objectives. The statistical connection provides intuition and interpretation for numerical results, and the applicability to multiple objectives allows system designers to accommodate different business requirements. Intriguingly, the relation between the optimal contract and the optimal statistical risk have the same functional form in both objectives (min-budget and min-pay). Moreover, multiplying optimal hypothesis tests by a constant whose value depends only on the statistical risk yields *approximately*-optimal contracts (Theorem 2).

Finally, we evaluate the empirical performance of cost-robust contracts by analyzing LLM evaluation benchmarks for two families of tasks. In the first experiment, we compare the performance of two-outcome contracts across code generation tasks with varying difficulty; results show that what determines the pricing scheme is the relative success rates of the models, not the task difficulty. In the second experiment, we compute multi-outcome contracts for an intricate conversational task evaluated via LLM-as-a-judge. Numerical results show that the optimal monotone cost-robust pricing scheme has an intuitive 3-level structure: pay nothing if the quality is poor, pay extra if it is exceptional, and pay a fixed baseline otherwise. We show our framework's flexibility by providing a comprehensive comparison across various contract objectives and simplicity constraints.

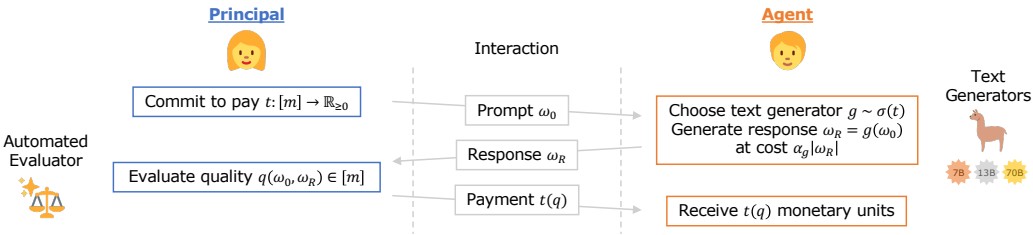

Figure 1: Interaction protocol. Principal commits to pay $t(q)$ monetary units according to response quality, and sends prompt $\omega_0$; Agent selects text generator $g \sim \sigma(t)$, and generates response $\omega_R = g(\omega_0)$ at cost $\alpha_g|\omega_R|$; Principal evaluates response quality $q(\omega_0, \omega_R)$, and pays accordingly.

## 1.1 Related work

Our main technical tool is algorithmic contract design (see [5, 20, 14] and subsequent works). Many works in this area address distributional robustness, e.g. [8], [14] which also studies approximation guarantees of simple contracts, and the recent [3] which presents a distributionally-robust contract design approach for delegation of learning. However, to our knowledge, none address cost-robustness. Connections between contract design and statistics have long been known to exist at a high level (see, e.g., [32]), and were recently explored by [6] in the context of adverse selection, and [31] for two-action min-budget contract. From a technical standpoint, our work is closest to [31], which only proves the statistical connection for the special case of two-action min-budget contracts. Finally, we note that our cost-robustness framework is general, and our characterization results may be of independent interest. Additional related work appears in Appendix A.

## 2 Problem Setting: Contract Design for Text Generation

We study the delegation of a text generation task from a strategic *principal* to *agent*, with a payment scheme designed to incentivize quality. Here we formulate the problem as a *contract design* instance.

### 2.1 Quality text generation (agent's perspective)

The core of our setting is a standard language generation task. Let $V$ be a vocabulary of tokens, and denote the set of all token sequences by $V^*$. A *text generator* $g : V^* \to V^*$ is a mapping from a textual prompt $\omega_0$ to a response $\omega_R$. We assume that prompts are sampled from a distribution $\omega_0 \sim D \in \Delta(V^*)$, and denote by $D_g$ the distribution of prompt-response pairs, where the prompt is sampled from $D$ and the response is generated by generator $g$. Given a prompt and generated response, a *quality evaluator* is a function $q : V^* \times V^* \to [m]$ which scores the response on a scale of $1, \ldots, m$. We use $F_g$ to denote the distribution over scores $[m]$ induced by applying the quality evaluator to a random pair $(\omega_0, \omega_R) \sim D_g$, and $F_{gj}$ to denote the probability of score $j \in [m]$.

The agent has access to a collection of possible text generators $\mathcal{G} = \{g_1, \ldots, g_n\}$, which we also refer to for convenience by their indices $[n]$. Each model $g_i \in \mathcal{G}$ is associated with a model-dependent cost $\alpha_i \geq 0$, which is the average cost (borne by the agent) of generating a single token from $g_i$. For convenience we write $D_i = D_{g_i}$ and $F_i = F_{g_i}$. Denote by $c_i = \alpha_i \mathbb{E}_{(\omega_0, \omega_R) \sim D_i}[\|\omega_R\|]$ the expected cost of using the $i$th generator. We assume w.l.o.g. that the costs are non-decreasing, i.e., $c_1 \leq \cdots \leq c_n$, and that they reflect the inherent quality of the models. In contract design terminology, the generators are the agent's possible *actions*. The agent can choose a single (*pure*) action, or a distribution over text generators $\sigma \in \Delta(\mathcal{G})$ known in game theory as a *mixed* action.[1] The cost $c_1$ of the lowest-cost action is the agent's "opportunity cost", and unless stated otherwise $c_1 = 0$.[2]

**As an abstract contract design problem.** The above setting is precisely a contract design setting with $n$ actions and $m$ outcomes [21]. Such a setting is defined by the pair $(F, c)$, where $F$ is an $n \times m$ matrix with distribution $F_i$ as its $i$th row for every $i$ (known as the *distribution matrix*), and where $c$ is a vector of costs. For every action $i$, $F_i$ and $c_i$ are the outcome distribution and cost, respectively.

---

[1]For example, the agent can generate responses using a larger model for 95% of requests, and apply the smaller model for the remaining 5%, corresponding to the mixed action $\sigma = (0.05, 0.95)$.

[2]Choosing the first action can be thought of as opting out of the task at cost $c_1$. If $c_1 = 0$ then the agent participates in the contract only if the expected utility is non-negative – a property known as *individual rationality*.

**Pay-for-performance and agent's utility.** To incentivize high quality text generation, the principal commits in advance to a pay-for-performance contract, which specifies the amount of payment to the agent for generating a response with a certain quality. More formally, given a quality evaluator $q$ with an output scale $1, \ldots, m$, a *contract* $t : [m] \to \mathbb{R}_{\geq 0}$ is a mapping from the estimated quality to the size of monetary transfer. Note that transfers are non-negative; this standard restriction is known as *limited liability*, and it mirrors the fact that when a principal hires an agent to perform a task, money flows in one way only (from principal to agent, and not vice versa). If transfers are increasing with score, we say $t$ is a *monotone* contract. Monotonicity is not without loss of generality, but is a desirable property as monotone contracts are generally simpler and easier to explain [14].

Given a contract $t \in \mathbb{R}_{\geq 0}^m$ and an action $\sigma \in \Delta(\mathcal{G})$, the agent's expected utility $u_A(t; \sigma)$ (a.k.a. the agent's profit) is the difference between the expected reward and the expected cost of text generation:

$$u_A(t; \sigma) = \mathbb{E}_{g_i \sim \sigma; (\omega_0, \omega_R) \sim D_i}[t(q(\omega_0, \omega_R)) - \alpha_i |\omega_R|] = \mathbb{E}_{g_i \sim \sigma; j \sim F_i}[t(j) - c_i],$$

where $(\omega_0, \omega_R) \sim D_i$ are the prompt and generated response, $t(q(\omega_0, \omega_R))$ is the payment transferred to the agent based on the quality of response, and $\alpha_i |\omega_R|$ is the agent's cost of generating the response. We assume the agent is rational and therefore selects, when facing contract $t$, an action $\sigma(t)$ which maximizes their expected profit (also known as the agent's *best response*):

$$\sigma(t) \in \arg\max_{\sigma \in \Delta(\mathcal{G})} u_A(t; \sigma).$$

As is standard in contract theory, we assume the agent breaks ties consistently and in a way that agrees with the principal's preferences.[3] The interaction model is summarized in Figure 1.

## 2.2 Designing the contract (principal's perspective)

We assume that the principal seeks to obtain text generated by the model $g_n \in \mathcal{G}$, the most advanced model with the (strictly) highest associated cost $c_n > c_{n-1}$. We refer to $g_n$ as the *target action*, i.e. the action which the principal wishes to incentivize. Taking the role of the principal, our goal is to design the "best" contract $t^*$ that incentivizes the agent to generate responses using the target model $g_n$. This is formalized by the following optimization problem:

$$t^* = \arg\min_{t \in \mathbb{R}_{\geq 0}^m} \|t\| \quad \text{s.t.} \quad \sigma(t) = \delta_{g_n}, \tag{1}$$

where $\|t\|$ is a norm of $t$ representing the principal's economic objective (see below), and $\delta_{g_n}$ is a point-mass distribution over text generators, supported by the target generator $g_n$. We denote the set of contracts incentivizing action $g_n$ by $\mathcal{T}(g_n) = \{t \in \mathbb{R}_{\geq 0}^m \mid \sigma(t) = \delta_{g_n}\}$, and further note that the assumption of a single target action serves as a foundational step towards more complex contract design scenarios (see Appendix B.1).

**Information structure (who knows what).** The agent's available actions $\mathcal{G}$ and the possible scores $[m]$ are known to both players. As the quality distributions $F_i$ can be learned from past data, we assume they are known to both principal and agent. As the costs of inference $\{\alpha_i\}$ depend on internal implementation details, we assume the costs are known to the agent but uncertain to the principal. We thus aim for a contract optimization framework which maximizes different types of objectives, and allows for optimization of $t$ even when the costs incurred by the agent are uncertain to the principal.

**Objectives: min-budget, min-pay and min-variance contracts.** In eq. (1), different norms $\|t\|$ correspond to different possible optimization goals of the principal: For example, a contract is *min-pay* if it incentivizes the target action using minimum total expected payment $\mathbb{E}_{j \sim F_n}[t(j)]$ among all contracts in $\mathcal{T}(g_n)$ [14]; In eq. (1), this corresponds to the $\ell_1$ norm weighted by the quality distribution of the target action. Similarly, a contract is *min-budget* if it incentivizes the target action using minimum budget $B_t = \max_j t(j)$ [31]; In eq. (1), this corresponds to the $\ell_\infty$ norm. Additionally, we also consider a natural *min-variance* objective, which was not previously studied to our knowledge. A min-variance contract minimizes the objective $\text{Var}(t)$, corresponding to a weighted $\ell_2$ norm. Optimal contracts for these objectives can be computed in polynomial time by solving a corresponding linear or convex-quadratic program (see Appendix D.1). We also consider approximately-optimal contracts:

**Definition 1** ($\eta$-optimal contract). *Let $\eta \geq 1$. For contract setting $(F, c)$, let $t^* \in \mathcal{T}(g_n)$ be the optimal contract with respect to objective $\|t\|$. A contract $t \in \mathcal{T}(g_n)$ is $\eta$-optimal if $\|t\| \leq \eta \|t^*\|$.*

---

[3]In our context, this means that if action $g_n$ is a best response for the agent, then the agent will choose $\sigma(t)$ that plays $g_n$ with probability 1 (see Section 2.2).

# 3 Hypothesis Testing and Contracts

This section sets the stage for connecting cost-robust contracts to statistical tests in Section 4.

## 3.1 Preliminaries

**Simple hypothesis tests** Consider two distributions $F_0, F_1 \in \Delta([m])$. Given $j \in [m]$ which is sampled from either $F_0$ or $F_1$, a *hypothesis test* is a function $\psi : [m] \to [0,1]$ which outputs 1 if $j$ is likely to have been sampled from $F_1$, and 0 otherwise[4]. In the hypothesis testing literature, $F_0$ is a *simple null hypothesis*, and $F_1$ is a *simple alternative hypothesis*. Performance measures of hypothesis tests are derived from the probabilities of making different types of errors: For a test $\psi$, the probability of false positives $\mathrm{FP} = \sum_{j=1}^m F_{0,j}\psi_j$ measures the rate of type-1 errors; This is when the test rejects the null hypothesis despite the sample being drawn from $F_0$. Similarly, the probability of false negatives $\mathrm{FN} = \sum_{j=1}^m F_{1,j}(1 - \psi_j)$ measures the rate of type-2 errors, i.e. when the test does not reject the null hypothesis despite the sample being drawn from $F_1$. We also denote the true positives by $\mathrm{TP} = \sum_{j=1}^m F_{1,j}\psi_j$; TP is also known as the test's power, and equal to $1 - \mathrm{FN}$.

**Composite hypothesis tests.** Consider now two *sets* of distributions $\{F_k\}_{k=1}^{n-1}, \{F_n\}$, where $F_i \in \Delta([m])$ for all $i \in [n]$. In hypothesis testing terms, $\{F_k\}_{k=1}^{n-1}$ is a *composite null hypothesis*. $\{F_n\}$ is a simple alternative hypothesis as before, and a composite hypothesis test $\psi$ outputs 1 if a given $j \in [m]$ is likely to have been sampled from $F_n$. To define performance in the composite case, we denote by $\mathrm{FP}_k = \sum_{j=1}^m F_{k,j}\psi_j$ the standard type-1 error between distributions $F_k$ and $F_n$. As the alternative hypothesis is still simple, definitions of FN and TP remain as before, using $F_n$ as the reference distribution. To measure the performance of hypothesis tests, it is common to take a *worst-case* approach, and define the composite FP as the standard type-1 error $\mathrm{FP}_k$ against the worst-case distribution $F_k$ in the null hypothesis set.

## 3.2 Risk and minimax tests

To formalize the notion of worst-case error, let $\psi$ be a composite hypothesis test for $\{F_k\}_{k=1}^{n-1}, \{F_n\}$. For any $k \in [n-1]$, define a *risk* function $r_k : [0,1]^m \to \mathbb{R}_{\geq 0}$ to be a mapping from $\psi$ to a risk score, treating $\psi$ as a simple hypothesis test between distributions $F_k$ and $F_n$. A natural way of measuring risk is by combining the test's two error types. One measure is the *sum* of errors, denoted by $R_k(\psi) := \mathrm{FP}_k + \mathrm{FN}$. A classic result by Neyman and Pearson shows that $R_k(\psi)$ is minimized by the *likelihood-ratio test* for any fixed $k$ [29]. Another measure is the *ratio* of false positives to true positives, denoted by $\rho_k(\psi) := \mathrm{FP}_k / \mathrm{TP}$. To generalize a risk measure $r_k$ to a composite hypothesis test, we adopt the worst-case approach and define $r(\psi) := \max_{k \in [n-1]}\{r_k(\psi)\}$. Thus, $R(\psi) = \max_{k \in [n-1]}\{R_k\} = \mathrm{FP} + \mathrm{FN}$, and $\rho(\psi) = \max_{k \in [n-1]}\{\rho_k\} = \mathrm{FP} / \mathrm{TP}$.

**Definition 2** (Minimax hypothesis test). *Let $\psi_F^*$ be a composite hypothesis test for $\{F_k\}_{k=1}^{n-1}, \{F_n\}$, and fix a risk function $r_k$. The test $\psi_F^*$ is* minimax optimal w.r.t. $r$ *if it minimizes the worst-case risk:*

$$\psi_F^* = \arg\min_{\psi \in [0,1]^m} \max_{k \in [n-1]} \{r_k(\psi)\} = \arg\min_{\psi \in [0,1]^m}\{r(\psi)\}.$$

*The* minimax sum-optimal test *and* minimax ratio-optimal test *are the minimax optimal tests with respect to the sum $R$ and the ratio $\rho$, respectively.*

Observe that the optimal risk of both types of tests is bounded by $r(\psi) \leq 1$, as the constant test $\psi_j = 0.5$ satisfies $R(\psi) = \rho(\psi) = 1$. We assume at least a small difference between the hypotheses, such that $r(\psi) < 1$. This allows us to define contracts based on these tests.

## 3.3 From tests to contracts and back

We derive "statistical" contracts from hypothesis tests by multiplying them by a function of the risk, and derive "contractual" tests from contracts by normalizing them: Consider a contract setting $(F, c)$, with either known costs and $b := c_n - c_1$, or a cost upper bound $b \geq c_n - c_1$. Fix a risk function $r$ and a corresponding budget function $B_r(\psi, b) \in \mathbb{R}$.

- **Test-to-contract**: Let $\psi$ be a test for sets $\{F_k\}_{k=1}^{n-1}, \{F_n\}$ with budget $B_r(\psi, b) \in [0, \infty)$. The corresponding *statistical contract* $t^{(r,\psi)}$ is $B_r(\psi, b) \cdot \psi$.

- **Contract-to-test**: Let $t$ be a contract. The corresponding *contractual test* $\psi^t$ is $t/\|t\|_\infty$.

---

[4]When $\psi(j)$ is fractional, we consider the output of the test to be 1 with probability $\psi(j)$, and 0 otherwise.

| Economic objective | Objective function | Statistical objective | Risk function |
|:---:|:---:|:---:|:---:|
| Min-budget | $\max_{j \in [m]} t_j$ | FP + FN | $\mathrm{Pr}_{F_k}(\psi = 1) + \mathrm{Pr}_{F_n}(\psi = 0)$ |
| Min-pay | $\mathbb{E}_{j \sim F_n}[t_j]$ | FP / TP | $\frac{\mathrm{Pr}_{F_k}(\psi=1)}{\mathrm{Pr}_{F_n}(\psi=1)}$ |

Table 1: Correspondence between cost-robust contracts and hypothesis tests, arising from Theorem 1.

We are interested in the following statistical contracts corresponding to the tests from Definition 2:

**Definition 3.** *Consider a contract setting $(F, c)$, with either known costs and $b := c_n - c_1$, or a cost upper bound $b \geq c_n - c_1$. The* sum-optimal statistical contract $B_R^*(b) \cdot \psi_R^*$ *is obtained from the minimax sum-optimal test $\psi_R^*$ multiplied by $B_R^*(b) := \frac{b}{1 - R(\psi_R^*)}$. The* ratio-optimal statistical contract $B_\rho^*(b) \cdot \psi_\rho^*$ *is obtained from the minimax ratio-optimal test $\psi_\rho^*$ multiplied by $B_\rho^*(b) := \frac{b}{\mathrm{TP}(\psi_\rho^*) - \mathrm{FP}(\psi_\rho^*)}$.*

## 4 Cost-Robust Contracts

In this section we state and prove our main result – a direct connection between composite hypothesis testing and cost-robust contracts. Consider a contract design setting $(F, c)$ with increasing costs $c_1 \leq \cdots \leq c_{n-1} < c_n$, where $n$ is the target action. In real-world settings, the principal may not have full knowledge of the agent's internal cost structure. We model this by assuming the principal is oblivious to the precise costs, but knows an upper bound $b \geq c_n - c_1$. We are interested in *robust* contracts that incentivize the target action for *any* cost vector compatible with the upper bound:

**Definition 4** (Cost-robust contracts). *Consider a distribution matrix $F$ and a bound $b > 0$ on the costs. Let $\mathcal{C}_b$ be an ambiguity set of all increasing cost vectors $c$ such that $c_n - c_1 \leq b$. A contract is $b$-cost-robust if it implements action $n$ for any cost vector $c \in \mathcal{C}_b$.*

Informally, our main theoretical result shows that optimal cost-robust contracts are optimal hypothesis tests up to scaling, where the scaler depends on the risk measure which the test optimizes. Our approach can be applied to several notions of optimality, and each optimality criterion for contracts corresponds to a different optimality criterion for hypothesis tests. Specifically, we derive the correspondence for min-budget and min-pay optimality of contracts. Formally (recall Definition 3):

**Theorem 1** (Optimal cost-robust contracts). *For every contract setting with distribution matrix $F$ and an upper bound $b$ on the (unknown) costs, let $\psi_R^*$ (resp., $\psi_\rho^*$) be the minimax sum-optimal (ratio-optimal) test with risk $R^*$ $(\rho^*)$ among all composite hypothesis tests for $\{F_k\}_{k=1}^{n-1}$, $\{F_n\}$. Then:*

- *The* sum-*optimal statistical contract $B_R^*(b) \cdot \psi_R^*$ is $b$-cost-robust with budget $b/(1 - R^*)$, and has the lowest budget among all $b$-cost-robust contracts.*

- *The* ratio-*optimal statistical contract $B_\rho^*(b) \cdot \psi_\rho^*$ is $b$-cost-robust with expected total payment $b/(1 - \rho^*)$, and has the lowest expected total payment among all $b$-cost-robust contracts.*

Table 1 summarizes the contract vs. test equivalences arising from Theorem 1. In the special case of contract design settings combining (i) a binary action space ($n = 2$), (ii) a zero-cost action ($c_1 = 0$), and (iii) a tight upper bound ($b = c_2$), the first half of Theorem 1 recovers a recently-discovered correspondence between hypothesis testing and (non-cost-robust) two-action min-budget contracts [31, Theorem 2]. Theorem 1 is more general since it applies to any number of actions as well as to the standard min-pay objective. Thus, Theorem 1 can also be seen as extending the interpretable format of optimal contracts for binary-action settings beyond two actions.

In Appendix D.2, we derive our main lemmas, which are used in Appendix D.4 to prove Theorem 1. Our proofs rely on two main assumptions. The first assumption is that the cost uncertainty bounds are known; as these bounds become looser, the required budget and expected payout increase linearly. The second assumption is that the contract design problem is implementable – i.e., that there exists some contract incentivizing the target action. In particular, implementability holds when text generated by the target model has the highest quality in expectation (see Appendix C.1). Generally, a contract design problem is implementable when the observed quality distribution of the target model can't be emulated by a combination of alternative models at a lower cost (see, e.g., [14]).

## 4.1 Additional properties of optimal cost-robust contracts

In this section, we focus for concreteness on min-*budget* cost-robust contracts, and establish their approximation guarantees as well as their functional form under structural assumptions. First, in analogy to [14], it is natural to examine the approximation guarantees of cost-robust contracts. We show (recall Definition 1):

**Theorem 2** (Approximation guarantees). *For every contract setting $(F, c)$, let $0 < a \leq b$ be a lower and upper bound on the difference between the target cost and any other cost, i.e., $(c_n - c_i) \in [a, b]$ for all $i \in [n-1]$. Then the min-budget $b$-cost-robust contract for $(F, c)$ is $\frac{b}{a}$-optimal with respect to the budget objective $\|t\|_\infty$, and the approximation ratio $\frac{b}{a}$ is tight.*

Proof in Appendix D.5. As a corollary, combining this result with Theorem 1 shows that statistical contracts are approximately optimal in the global sense: For any contract setting $(F, c)$ with corresponding minimax sum-optimal hypothesis test $\psi_R^*$, the contract $t = \frac{c_n - c_1}{1 - R^*} \psi_R^*$ is $\eta$-optimal with respect to the budget metric and $\eta = \frac{c_n - c_1}{c_n - c_{n-1}}$. We also note that similar results hold for min-*pay* cost-robust contracts.

We next turn to consider the functional form of optimal cost-robust contracts (i.e., why their payments are as they are). One of the criticisms of optimal (non-robust) contracts is that the payments seem arbitrary and opaque. Compared to this, cost-robust contracts are more transparent and explainable. We show two additional results regarding their format, leveraging the connection to minimax hypothesis testing.

The first result explains the budget: By the minimax principle, there is a "least favorable distribution" (to use terminology from statistics) or, equivalently, a mixed strategy over the rows of $F$ (to use terminology from game theory) such that no test can achieve for it better risk than the minimax risk $R^*$. We show that the budget of the optimal cost-robust contract can be interpreted using this distribution. Formally, let $\|F_1 - F_2\|_{\text{TV}}$ be the total variation distance between distributions $F_1, F_2$, then the budget is as follows (see Appendix D.5 for a proof):

**Proposition 1** (Distribution distance determines budget). *For every contract setting with distribution matrix $F$ and spread of costs $c_n - c_1 \leq b$, the minimum budget of a $b$-cost-robust contract is $\max_{\lambda \in \Delta([n-1])} b / \left\| F_n - \sum_{i<n} \lambda_i F_i \right\|_{\text{TV}}$.*

The distribution $\sum_{i<n} \lambda_i F_i$ that maximizes the above expression is the least favorable one. Intuitively, the closer it is to the target distribution $F_n$, the larger the budget needed for the agent to distinguish among them and prefer the target action. We also note that the required budget approaches infinity when the worst-case distribution distance approaches zero, coinciding with the implementability characterization known from the literature (see Appendix C.1). The second set of results adds standard structure to the distribution matrix $F$ to obtain even simpler contract formats:

**Definition 5** (Monotone Likelihood Ratio (MLR)). *A distribution matrix $F$ satisfies MLR if $F_{i,j}/F_{i',j}$ is monotonically increasing in $j \in [m]$ for all $i > i'$.*

Intuitively, if $F$ satisfies MLR, then the higher the outcome $j$, the more likely it is to origin from a more costly distribution $F_i$ than from $F_{i'}$ (recall that costs $c_i$ are increasing in $i$). Consider minimax composite hypothesis tests for $\{F_k\}_{k=1}^{n-1}, \{F_n\}$; if $F$ does not satisfy MLR, optimal such tests may require randomization (i.e., $\psi_j \in (0, 1)$ for some outcome $j$) and/or non-monotonicity (i.e. $\psi_j > \psi_{j'}$ for some pair of outcomes $j < j'$). However, if MLR holds for $F$, then nice properties (determinism, monotonicity) hold for minimax tests (e.g., by the Karlin-Rubin theorem [29]), and consequently also for cost-robust contracts (see Appendix D.6 for a proof):

**Proposition 2** (MLR induces threshold simplicity). *For every contract setting with distribution matrix $F$ that satisfies MLR, and with spread of costs $c_n - c_1 \leq b$, the min-budget $b$-cost-robust contract for $F$ is a monotone threshold contract, which pays full budget to the agent for every outcome $j$ above some threshold $j^*$.*

For min-pay rather than min-budget, under MLR we get a monotone contract with a single positive payment, which is optimal (see [14, Lemma 7]). Finally, we note that finding cost-robust min-budget contracts with two levels of pay ("all-or-nothing" [31]) is computationally hard in the general case, as the reduction by [31, Theorem 3] also applies in the cost-robust case (see Appendix D.8).

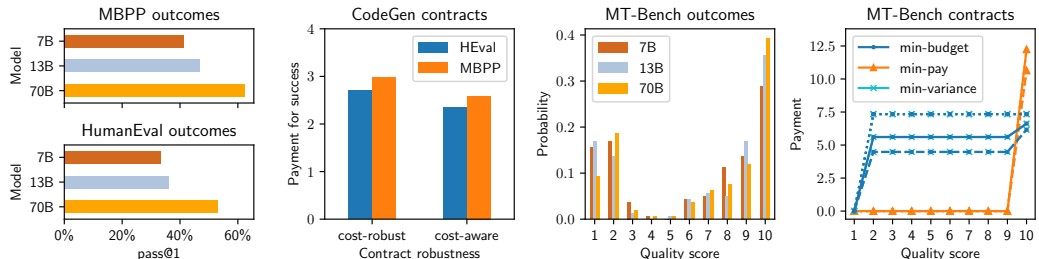

Figure 2: Empirical evaluation results. (**Left**) Outcome distributions and optimal contracts for Code Generation data, Section 5.1. (**Right**) Outcome distributions and optimal contracts for MT-Bench data, Section 5.2. For the contracts plot, solid lines represent cost-robust contracts, dashed lines represent cost-aware contracts, and dotted lines represent threshold contracts.

## 5  Empirical Evaluation

We evaluate the empirical performance of our cost-robust contracts using LLM evaluation benchmarks. We compute binary and multi-outcome contracts for two distinct families of tasks based on evaluation scores from known benchmark datasets, optimizing the contract objectives set forth in Section 2.2. Our action space consists of the 7B, 13B, and 70B parameter model versions of the open-source Llama2 and CodeLlama LLMs [37, 30], which share the same architecture and hence similar inference costs. The benchmark data is used to create an empirical outcome distribution for each LLM in the action space. In both cases, contract optimization targets of the largest model variant (70B), which is the most performant and costly. Implementation details are provided in Appendix E.3, and code is available at: `https://github.com/edensaig/llm-contracts`.

**Cost estimation.**  To estimate the inference costs of the language models, we leverage their open-source availability. We use energy consumption data from the popular Hugging Face LLM-Performance Leaderboard [22, 23], which we then convert to dollar values using conservative cost estimates (see Appendix E.1). As a first-order assumption of cost uncertainty, we assume that inference costs of alternative generators are bounded from below by the cost of the most energy-efficient alternative model ($c_1$), and bounded from above by the cost of the alternative model with the highest energy consumption ($c_{n-1}$).

### 5.1  Binary-outcome contracts across tasks (code generation)

We begin by analyzing a simple contract design setting across benchmarks of varying difficulties. We use the LLM task of code-generation which has $m = 2$ outcomes: pass or fail. The analysis of a binary outcome space is motivated by the following theoretical property:

**Proposition 3.** *For any contract design problem with $m = 2$ where the most-costly (target) action has the highest pass rate, the optimal contract is identical for all optimality objectives (min-pay, min-budget, and min-variance). Moreover, the optimal contract satisfies $t_{\text{pass}} > 0$, $t_{\text{fail}} = 0$.*

Proof in Appendix D.7. Proposition 3 allows us to compare performance across different evaluation tasks without being sensitive to the choice of contract objective and constraints such as monotonicity.

**Datasets.**  We use evaluation data from two distinct benchmarks, which represent differing degrees of task difficulty. The Mostly Basic Programming Problems (MBPP) benchmark [4] contains 974 entry-level programming problems with 3 unit tests per problem. The HumanEval benchmark [11] consists of 164 hand-written functional programming problems. Included in each programming problem is a function signature, a doc-string prompt, and unit tests. There is an average of 7.7 unit tests per problem, and the overall score is based on a pass/fail evaluation of the responses. For each of these benchmarks, we create a binary-outcome ($m = 2$) contract from the pass rates of the CodeLlama model family (`CodeLlama-{7B,13B,70B}`). We use the pass@1 values from the CodeLlama paper [30] (success rates for a single response), as they capture a setting where the agent gets paid for each response.

|  | Cost-aware | | | Cost-robust | | |
| --- | --- | --- | --- | --- | --- | --- |
|  | $\mathbb{E}[t]$ | Budget | stdev$(t)$ | $\mathbb{E}[t]$ | Budget | stdev$(t)$ |
| Min-Pay | **4.19** | 10.6 | 5.2 | 4.82 (+15%) | 12.2 | 5.98 |
| Min-Budget | 4.73 | **6.16** | 1.71 | 5.48 | 6.63 (+1.7%) | 1.83 |
| Min-Variance | 4.73 | 6.16 | **1.71** | 5.48 | 6.63 | 1.83 (+6.7%) |

Table 2: Monotone contracts curated on the MT-bench dataset. Costs are in units of \$/1M tokens, and are order-of-magnitude estimates based on typical prices of electricity (see Appendix E.1). The numbers in bold denote the relative increase from the optimal monotone contract that the cost-robustness sacrifices in each setting. The percentages denote the price of cost-robustness: how much the objective values increase relative to the cost-aware setting.

**Task difficulty and optimal pay.** Figure 2 (Left-Center) presents optimal cost-aware and cost-robust contracts for code-generation. We observe that cost uncertainty entails a consistent increase in payment across tasks: For MBPP, we observe a 14.9% increase, and for HumanEval we observe a similar 14.7% increase. Additionally, while the MBPP task is easier than HumanEval (i.e. characterized by higher pass rates), the resulting contracts for MBPP are more expensive. This demonstrates the fundamental connection between contracts and statistics: In MBPP, there is smaller gap between the performance of the target model (70B) and the performance of the alternatives. This makes the highest-performing model harder to detect, increasing the cost of the contract. The required payments in this case thus depend on the absolute differences between pass rates, rather than absolute values.

## 5.2 Multi-outcome contracts

To understand the relation between different optimality objectives and constraints, we analyze optimal contracts in an expressive multi-outcome ($m = 10$) environment based on MT-Bench.

**Dataset.** The MT-Bench benchmark [41] is designed to evaluate the conversational and instruction-following abilities of LLMs in multi-turn (MT) conversational settings. The benchmark consists of 80 prompts in the format of multi-turn questions, and the evaluation dataset includes LLM-as-a-judge evaluations on (prompt,response) pairs from various models, using GPT-4 as the judge. In the dataset, each (prompt,response) pair is given discrete *response quality* scores in the range $1, \ldots, 10$. These scores define our contract outcome space. In consistence with the analysis in section 5.1, we use the outcome distributions of (Llama-2-{7B,13B,70B}-chat), and target the 70B model. Outcome distributions for the MT-Bench dataset are presented in Figure 2.

**Simple optimal contracts.** In practical applications, contracts with a simple functional form are often preferred since they are easier to comprehend. We compute optimal contracts with two types of simplicity constraints: monotone contracts (weakly increasing payout), and threshold contracts (full budget for all scores above a threshold, and zero otherwise). Results are presented in Figure 2 (Right), and Table 2. For the min-budget and min-variance criteria, monotone cost-robust contracts have an intuitive three-level structure: Zero pay for outputs with the lowest quality score, base pay for intermediate scores, and extra pay for outputs with the highest quality score. While threshold contracts may resemble current pricing schemes more closely (see Appendix E.1), monotone contracts enable a lower overall budget while still maintaining a simple functional form. For min-pay, monotone cost-robust contracts are in themselves threshold contracts, however they may deter risk-averse agents as they only pay for highest-quality outputs. In Appendix E.2, we additionally analyze non-monotone contracts, and show that further economic efficiency can be achieved by sacrificing simplicity.

**Price of cost-robustness.** Table 2 compares cost-robust and cost-aware monotone contracts across different performance metrics. We observe that cost-robust contracts setting sacrifice a marginal increase in objective values: a $15\%$ increase in the min-pay objective, an $1.7\%$ increase if optimizing for budget, and $6.7\%$ increase when optimizing for minimum variance. We refer to Appendix E.2 for further analysis of cost-robustness in the non-monotone setting.

# 6 Discussion

In this paper, we introduce cost-robust contracts as a means to address the emerging problem of moral hazards in LLM inference. Our aim is to offer flexible payment schemes that ensure integrity in current LLM markets, even when facing challenges of incomplete information. One of the key insights from our study is that cost-robust contracts can be relevant and effective in practical settings. Moreover, we generalize the work paved by Saig et al. [31] by uncovering stronger connections between the fields of contract design and statistical hypothesis testing. These connections underscore the statistical intuition that is prevalent in contract design.

Despite the promising results, our work still has several limitations that would do well to be addressed in future research. For one, the data we capture through the evaluation benchmarks does not accurately reflect real-world distributions, where the prompt space is much richer. A natural direction for future work is to explore approximation guarantees when learning contracts from data. Additionally, our analysis relies on a set of assumptions regarding the cost uncertainty and estimations, which should be carefully considered when designing contracts for Generative AI. Lastly, it would also be interesting to see our contract design framework applied to markets with a more elaborate action space.

**Acknowledgements.** The authors would like to thank Nir Rosenfeld, Ariel Procaccia, Stephen Bates, and Michael Toker for their insightful remarks and valuable suggestions. Eden Saig is supported by the Israel Council for Higher Education PBC scholarship for Ph.D. students in data science. This work received funding from the European Research Council (ERC) under the European Union's Horizon 2020 research and innovation program (grant No.: 101077862, project: ALGOCONTRACT, PI: Inbal Talgam-Cohen), by the Israel Science Foundation (grant No.: 3331/24), by the NSF-BSF (grant No.: 2021680), and by a Google Research Scholar Award.

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

## A  Additional Related Work

**Detection.**  As a possible alternative to a contract-design approach, the LLM content detection literature develops tools which attempt to detect machine-generated text, and distinguish between different text generators [38, 26, 40, 39]. Using such tools, a principal could deploy an LLM content detector and penalize firms who are not labeled as using target text generator. From this perspective, contract design is a complementary approach which provides guidelines for positive incentives in case a generated text gets accepted, an approach considered more effective at encouraging participation. Additionally, our pay-for-performance framework supports richer outcomes spaces beyond binary pass/fail, enabling more granular, and thus more efficient, control of incentives.

**AGT and LLMs.**  On a broader perspective, our work further promotes the role of Algorithmic Game Theory in the economics of Generative AI. Previous works include: Duetting et al. [13] who design auctions that merge outputs from multiple LLMs; Harris et al. [18] who offer a Bayesian Persuasion setting where the sender can use Generative AI to simulate the receiver's behavior; and Fish et al. [15] who leverage the creative nature of LLMs to enhance social choice settings.

## B  Extensions

### B.1  Targeting a set of high-quality models

For high-stake tasks such as summarizing medical information, it makes sense to target the most advanced model. In other scenarios, specifying a single target action can act as an intermediate step toward the final contract design.

For instance, a principal might aim to incentivize the use of any model that meets a certain quality threshold, e.g., requiring text generation from any LLM with more than 70B parameters. Formally, given a set of models $\mathcal{G} = \{g_1, \ldots, g_n\}$ with associated costs $c_1 < c_2 < \cdots < c_n$, we assume that higher-cost models offer higher quality. Let $k \in [n]$ represent the index of the minimum quality model that the principal seeks to target, such that the goal is to incentivize any model $g_i \in \{g_k, \ldots, g_n\}$.

To compute the optimal contract in this setting, the principal enumerates over the different target models $g_i \in \{g_k, \ldots, g_n\}$, designs an optimal single-target contract for each, and selects the "best" contract among the resulting designs – formally, the contract minimizing $\|t\|$ for the appropriate norm (see Section 2.2). Since all enumerated contracts are designed to satisfy incentive compatibility and potentially cost-robustness, the optimal contract among them also satisfies these properties.

## C  Contract Implementability

### C.1  Conditions for implementablity

The implementatbility of min-pay contracts is discussed in [14, Appendix A.2], and the implementatbility of min-budget contracts is discussed in [31, Appendix B.3.1]. In both cases, the implementability of the contract design problems is characterized by the following condition:

**Proposition 4** (Implementablity; [e.g., 14, Proposition 2])**.** *In a contract design problem $(F, c)$ with $n$ possible actions, an action $i \in [n]$ is implementable if and only if there is no convex combination of alternative actions that results in the same outcome distribution $\sum_{i' \neq i} \lambda_{i'} F_{i'} = F_i$ but lower cost $\sum_{i' \neq i} \lambda_{i'} c_{i'} < c_i$.*

We note that Proposition 4 holds for all objectives $\|t\|$ described in eq. (1), as feasibility only depends on the incentive compatibility constraints. Adding to the result above, we show that implementability holds in the the intuitive case where the target model has the highest expected quality:

**Proposition 5** (Highest expected quality implies implementability)**.** *For a contract design problem $(F, c)$, denote the expected quality of action $i$ by $q_i = \mathbb{E}_{j \in F_i}[j]$. If $q_i < q_n$ for all $i < n$, then the contract is implementable.*

*Proof.* By contradiction. Assume that $q_i < q_n$ for all $i < n$ but the contract design problem is not implementable. By Proposition 4, there exists a convex combination of actions such that $\sum_{i<n} \lambda_i F_i = F_n$ and $\sum_i c_i < c_n$. Denote the convex combination of distributions by $F_\lambda$. By definition, it holds that:

$$\mathbb{E}_{j \sim F_\lambda}[j] = \sum_{i<n} \lambda_i \mathbb{E}_{j \sim F_i}[j] = \sum_{i<n} \lambda_i q_i$$

But from the infeasibility assumption $\sum_{i<n} \lambda_i F_i = F_n$, and therefore it also holds that:

$$\mathbb{E}_{j \sim F_\lambda}[j] = \mathbb{E}_{j \sim F_n}[j] = q_n$$

and therefore $\sum_{i<n} \lambda_i q_i < q_n$, which contradicts the initial assumption that $q_i < q_n$ for all $i$. $\qquad\square$

## C.2 Designing cost-robust contracts for strictly-intermediate target actions

Our main analysis technique for cost-robust contracts requires a separation between the interval covering the costs of target actions, and the interval covering the costs of alternative actions (see proofs of Lemma 2 and Lemma 3). This assumption does not hold when the principal targets a strictly intermediate model, formally $g_i \in \mathcal{G}$ such that $i < n$ and $c_i < c_n$. This captures scenarios where the principal seeks to incentivize text generation from a medium-sized model, but not from larger, costlier models that are available. In such instances, guaranteeing the implementability of cost-robust contracts requires additional assumptions.

To illustrate this, consider the following design setting, with $n = 3$ actions and $m = 2$ outcomes:

$$
\begin{aligned}
F_1 &= (1, 0) & c_1 &= 0 \\
F_2 &= (0, 1) & c_2 &= 1 \\
F_3 &= (0.5, 0.5) & c_3 &= 10
\end{aligned}
$$

Assume that the principal targets action $i^* = 2$. In this setting, the cost difference is upper-bounded by $c_{i^*} - c_i \leq c_2 - c_1 = 1$, and the cost-robust min-budget contract corresponding to the bound $b = 1$ is given by Theorem 1:

$$t = \left(0, \frac{b}{\|F_3 - F_2\|_{\mathrm{TV}}}\right) = \left(0, \frac{1}{0.5}\right) = (0, 2)$$

This contract is incentive compatible with respect to the target action, yielding expected utilities $u_A(t, 1) = 0$ for action 1, $u_A(t, 2) = 1$ for action 2, and $u_A(t, 3) = -9$ for action 3.

However, consider the following setting, with identical target and different outcome distributions:

$$
\begin{aligned}
F_1 &= (1, 0) & c_1 &= 0 \\
F_2 &= (0.5, 0.5) & c_2 &= 1 \\
F_3 &= (0, 1) & c_3 &= 10
\end{aligned}
$$

In this setting, an incentive-compatible contract exists (e.g., $t = (0, 4)$), and the same cost upper bound $b = 1$ holds, however the cost-robust contract design problems are unfeasible by Proposition 4, as in the truncated contract design setting $(F = (F_1, F_2, F_3), c^{\mathrm{const}} = (0, 1, 0))$ there exists a convex combination of alternative actions $\lambda_1 = 0.5, \lambda_3 = 0.5$ which satisfies $\sum_{i \in \{1,3\}} \lambda_i F_i = F_2$ and $\sum_{i \in \{1,3\}} \lambda_i c_i^{\mathrm{const}} = 0 < c_2 = 1$.

## D Deferred Proofs

### D.1 Convex programs and equivalent forms

In this section, we include linear programs (LPs) for optimizing contracts and hypothesis tests. Non-negativity constraints on the variables are ommitted where clear from context.

By definition (see Section 2.2), a contract $t$ is min-budget with respect to target action $i$ if and only if it is an optimal solution to the following MIN-BUDGET LP, where IC stands for incentive compatibility (i.e., the constraints that ensure the agent's best response to $t$ is choosing action $i$):

$$
\begin{aligned}
\min_{t \in \mathbb{R}_{\geq 0}^m, B \in \mathbb{R}_{\geq 0}} \quad & B \\
\text{s.t.} \quad & \sum_j F_{i'j} t_j - c_{i'} \leq \sum_j F_{ij} t_j - c_i && \forall i' \neq i \quad \text{(IC)} \quad\quad (2) \\
& t_j \leq B && \forall j \in [m] \quad \text{(BUDGET)}
\end{aligned}
$$

**Proposition 6** (Equivalent form to the MIN-BUDGET LP; [31, B.2]). *When eq. (2) is feasible, the variable transformation $(t, B) \mapsto (\psi/\beta, 1/\beta)$ yields an equivalent LP which we refer to as the statistical LP:*

$$\max_{\psi \in [0,1]^m, \beta \in \mathbb{R}_{\geq 0}} \quad \beta$$
$$\text{s.t.} \quad \sum_j F_{i'j}\psi_j + \sum_j F_{ij}(1 - \psi_j) \leq 1 - (c_i - c_{i'})\beta \qquad \forall i' \neq i \tag{3}$$

Similarly to the MIN-BUDGET LP, we have the MIN-PAY LP:

$$\min_{t \in \mathbb{R}_{\geq 0}^m} \quad \sum_j F_{ij}t_j$$
$$\text{s.t.} \quad \sum_j F_{i'j}t_j - c_{i'} \leq \sum_j F_{ij}t_j - c_i \qquad \forall i' \neq i \quad \text{(IC)} \tag{4}$$

There are also natural LP formulations for hypothesis testing. A hypothesis test $\psi$ is the minimax sum-optimal test w.r.t. risk (see Definition 2) if and only if it is an optimal solution to the following LP:

$$\min_{\psi \in [0,1]^m, r \in \mathbb{R}_{\geq 0}} \quad r$$
$$\text{s.t.} \quad \sum_j F_{ij}\psi_j + \sum_j F_{nj}(1 - \psi_j) \leq r \qquad \forall i < n \tag{5}$$

**Proposition 7** (Dual of statistical LP). *The dual of eq. (3) is given by:*

$$\min_{\lambda \in \mathbb{R}_{\geq 0}^{n-1}, \mu \in \mathbb{R}_{\geq 0}^m} \quad \sum_{j \in [m]} \mu_j$$
$$\text{s.t.} \quad \sum_{i < n} (F_{n,j} - F_{i,j})\lambda_i \leq \mu_j \qquad \forall j \in [m]$$
$$\sum_{i < n} (c_n - c_i)\lambda_i \geq 1$$
$$\mu_j \geq 0 \qquad\qquad\qquad \forall j \in [m]$$
$$\lambda_i \geq 0 \qquad\qquad\qquad \forall i \in [n-1] \tag{6}$$

*Proof.* Denote:
$$x = (\psi_1, \ldots, \psi_m, \beta) \in \mathbb{R}^{m+1}.$$

$$A = \left( \begin{array}{ccccc|c}
F_{1,1} - F_{n,1} & & \cdots & & F_{1,m} - F_{n,m} & c_n - c_1 \\
 & \ddots & & & & \vdots \\
\vdots & & F_{i,j} - F_{n,j} & & \vdots & c_n - c_i \\
 & & & \ddots & & \vdots \\
F_{(n-1),1} - F_{n,1} & & \cdots & & F_{(n-1),m} - F_{n,m} & c_n - c_{n-1} \\
\hline
 & & & & & 0 \\
 & & I_{m \times m} & & & \vdots \\
 & & & & & 0
\end{array} \right)$$

$$b = (\underbrace{0, \ldots, 0}_{n-1 \text{ times}}, \underbrace{1, \ldots, 1}_{m \text{ times}}) \in \mathbb{R}^{n-1+m}.$$

$$c = (\underbrace{0, \ldots, 0}_{m \text{ times}}, 1) \in \mathbb{R}^{m+1}.$$

Using these notations, eq. (3) can be written as:

$$\max_{x \geq 0} \quad c^T x$$
$$\text{s.t.} \quad Ax \leq b \tag{7}$$

The symmetric dual LP of eq. (7) is given by:

$$\min_{y \geq 0} \quad b^T y$$
$$\text{s.t.} \quad A^T y \geq c \tag{8}$$

Denote:

$$y = (\lambda_1, \ldots, \lambda_{n-1}, \mu_1, \ldots, \mu_m).$$

Unpacking the matrix notations in eq. (8) yields the following:

$$\min_{\lambda \in \mathbb{R}^{n-1}_{\geq 0}, \mu \in \mathbb{R}^m_{\geq 0}} \quad \sum_{j \in [m]} \mu_j$$
$$\text{s.t.} \quad \sum_{i < n} (F_{i,j} - F_{n,j}) \lambda_i + \mu_j \geq 0 \qquad \forall j \in [m]$$
$$\sum_{i < n} (c_n - c_i) \lambda_i \geq 1$$

which is equivalent to eq. (6). $\qquad \square$

**Proposition 8.** *A min-variance contract is an optimal solution for the following quadratic program:*

$$\min_{t \in \mathbb{R}^m_{\geq 0}} \quad t^T V t$$
$$\text{s.t.} \quad \sum_j F_{i'j} t_j - c_{i'} \leq \sum_j F_{ij} t_j - c_i \qquad \forall i' \neq i \quad \text{(IC)} \tag{9}$$

*Where $V$ is a positive semi-definite matrix depending on the target action distribution $F_i$.*

*Proof.* Denote the contract by $t \in \mathbb{R}^m_{\geq 0}$ and the probability distribution of the target action by $p \in \Delta([m])$. We use the following matrix notations:

$$\mathbf{p} = \begin{pmatrix} p_1 \\ \vdots \\ p_m \end{pmatrix}; \quad \mathbf{t} = \begin{pmatrix} t_1 \\ \vdots \\ t_m \end{pmatrix}; \quad \mathbf{1} = \begin{pmatrix} 1 \\ \vdots \\ 1 \end{pmatrix}; \quad \text{diag}(\mathbf{p}) = \begin{pmatrix} p_1 & & \\ & \ddots & \\ & & p_m \end{pmatrix}$$

The variance of $t$ is given by:

$$\text{Var}(t) = \mathbb{E}_{j \sim p} \left[ (t_j - \mathbb{E}[t])^2 \right]$$
$$= \sum_j p_j \left( t_j - \sum_k p_k t_k \right)^2$$
$$= \text{diag}(\mathbf{p}) \left\| I\mathbf{t} - \mathbf{1}\mathbf{p}^T \mathbf{t} \right\|^2$$
$$= \left\| \text{diag}(\sqrt{\mathbf{p}}) \left( I - \mathbf{1}\mathbf{p}^T \right) \mathbf{t} \right\|^2$$

Denote $R = \text{diag}(\sqrt{\mathbf{p}}) (I - \mathbf{1}\mathbf{p}^T)$, and $V = R^T R$. Then:

$$\text{Var}(t) = \| Rt \|$$
$$= t^T R^T R t$$
$$= t^T V t$$

Note that $V = R^T R$ is a Gram matrix. It is therefore positive semi-definite, and the quadratic program is convex. $\qquad \square$

## D.2 Main lemmas

The next lemmas are the workhorses of our theoretical results. We use $\mathcal{T}_{(F,c)}$ to denote the set of contracts incentivizing the target action $n$ in a contract design setting $(F, c)$; the contracts in $\mathcal{T}_{(F,c)}$ are also known as the *feasible solutions* of the setting. For simplicity we focus on settings for which the set of feasible solutions is nonempty (i.e., the target action is implementable). Given either a non-decreasing cost vector $c = (c_1, \ldots, c_{n-1}, c_n) \in \mathbb{R}^n_{\geq 0}$ and an index $k \in [n-1]$, or a cost $c_n$ and a constant $c' < c_n$, define

$$c^{(k)} := (c_k, \ldots, c_k, c_n) \in \mathbb{R}^n_{\geq 0}; \quad c^{\text{const}} := (c', \ldots, c', c_n).$$

These are vectors with uniform costs (up to $c_n$). Note that the costs in $c^{(1)}$ are (weakly) lower than those in $c$, and vice versa for $c^{(n-1)}$:

$$c^{(1)} \leq c \leq c^{(n-1)}$$

Where the relation $\leq$ is defined element-wise. Intuitively, since the agent gravitates towards lower costs, it is harder to incentivize the target action against lower costs. We formalize this as follows:

**Lemma 1** (Incentivizing against lower costs is harder). *Consider a distribution matrix $F$, and two cost vectors $c \leq \bar{c} \in \mathbb{R}^n_{\geq 0}$ satisfying $c_n = \bar{c}_n$ (i.e., $c$ is dominated by $\bar{c}$, and the cost of the target action coincides). Then the sets of feasible solutions for contract design settings $(F, c)$ and $(F, \bar{c})$ satisfy $\mathcal{T}_{(F,c)} \subseteq \mathcal{T}_{(F,\bar{c})}$.*

**Corollary 1.** *For every contract design setting $(F, c)$, the set $\mathcal{T}_{(F,c)}$ of feasible solutions satisfies $\mathcal{T}_{(F,c^{(1)})} \subseteq \mathcal{T}_{(F,c)} \subseteq \mathcal{T}_{(F,c^{(n-1)})}$.*

Consider now a contract setting $(F, c^{\text{const}})$, where the action costs are uniformly equal to $c'$ except for the target action (which is more costly). We show that for such a setting, the optimal contract for incentivizing the target action has an interpretable format closely related to hypothesis testing. Recall the notions of sum-optimal and ratio-optimal statistical contracts from Definition 3; then:

**Lemma 2** (Min-budget optimality in uniform-cost settings). *For every contract design setting $(F, c^{\text{const}})$, the min-budget contract coincides with the sum-optimal statistical contract $B_R^*(c_n - c') \cdot \psi_R^*$, and the optimal budget is $(c_n - c')/(1 - R^*)$.*

**Lemma 3** (Min-pay optimality in uniform-cost settings). *For every contract design setting $(F, c^{\text{const}})$, the min-pay contract coincides with the ratio-optimal statistical contract $B_\rho^*(c_n - c') \cdot \psi_\rho^*$, and the optimal expected total payment is $(c_n - c')/(1 - \rho^*)$.*

Proofs appear in Appendix D.3, establishing also the other direction:

**Observation 1.** *Let $(F, c^{\text{const}})$ be a contract design setting. Then the minimax sum-optimal test among the composite hypothesis tests for $\{F_k\}_{k=1}^{n-1}$, $\{F_n\}$ is obtained by normalizing the min-budget contract, and the minimax ratio-optimal test is obtained by normalizing the min-pay contract.*

## D.3 Proofs of main lemmas

*Proof of Lemma 1.* For target action $n$ and any alternative action $i < n$, the (IC) constraint of the MIN-BUDGET LP (eq. (2)) is given by:

$$\sum_j F_{ij} t_j - c_i \leq \sum_j F_{nj} t_j - c_n.$$

Rearranging the terms yields:

$$\sum_j (F_{ij} - F_{nj}) t_j \leq c_i - c_n. \tag{10}$$

Let $\bar{c}$ be a cost vector satisfying $c \leq \bar{c}$ and $c_n = \bar{c}_n$. The costs $c_k$ are assumed to be increasing in $k$, and therefore $\bar{c}_i - \bar{c}_n = \bar{c}_i - c_n < 0$ for all $i$. Moreover, as $c_i \leq \bar{c}_i$ for all $i < n$, the RHS of eq. (10) satisfies:

$$\underbrace{c_i - c_n}_{\text{RHS of } (F,c) \text{ LP}} \quad \leq \quad \underbrace{\bar{c}_i - \bar{c}_n}_{\text{RHS of } (F,\bar{c}) \text{ LP}} \quad < 0.$$

Hence, the (IC) constraints of the $(F, c)$ contract design problem are more restrictive than the (IC) constraints of the $(F, \bar{c})$ design problem. Since the design problems $(F, c)$, $(F, \bar{c})$, only differ in the RHS of the (IC) constraints, the sets of feasible solutions satisfy the desired inclusion relation. $\quad \square$

*Proof of Lemma 2.* Under the "statistical" variable transformation $(t, B) \mapsto (\psi/\beta, 1/\beta)$, the MIN-BUDGET LP for $(F, c^{\mathrm{const}})$ is given by eq. (3):

$$\max_{\psi \in [0,1]^m, \beta \in \mathbb{R}_{\geq 0}} \quad \beta$$
$$\text{s.t.} \quad \sum_j F_{ij}\psi_j + \sum_j F_{nj}(1 - \psi_j) \leq 1 - (c_n - c')\beta \qquad \forall i < n$$

Applying the variable transformation $r = 1 - (c_n - c')\beta$ yields the following equivalent LP:

$$\min_{\psi \in [0,1]^m, r \in \mathbb{R}_{\geq 0}} \quad r$$
$$\text{s.t.} \quad \sum_j F_{ij}\psi_j + \sum_j F_{nj}(1 - \psi_j) \leq r \qquad \forall i < n$$

This LP is equivalent to the minimax sum-optimal test $\psi_R^*$ in eq. (5), and therefore the optimal solution $\psi^*$ is precisely this test. By the same equivalence, the optimal value of the optimization parameter $r$ satisfies $r^* = R^*$, where $R^*$ is the minimax risk of the testing problem (i.e., the risk of $\psi_R^*$). By construction, the optimal $\beta$ satisfies $\beta^* = \frac{1-r^*}{c_n - c'} = \frac{1-R^*}{c_n - c'}$, and therefore the minimal budget is $B^* = \frac{1}{\beta^*} = \frac{c_n - c'}{1 - R^*}$ which in the notation of Definition 3 is $B_R^*(c_n - c')$. Reversing the variable transformation we get $t^* = \psi^*/\beta^* = B^* \cdot \psi^*$, which is equal to the sum-optimal statistical contract $B_R^*(c_n - c') \cdot \psi_R^*$, as required. $\qquad \square$

*Proof of Lemma 3.* For the min-pay contract design problem, introduce an auxiliary variable $\beta > 0$, and define a "statistical" variable transformation $t \mapsto \frac{c_n - c'}{\beta}\psi$, where $\psi \in [0,1]^m$. The MIN-PAY LP (eq. (4)) transforms into:

$$\min_{\psi \in [0,1]^m, \beta > 0} \quad \frac{c_n - c'}{\beta} \sum_j F_{nj}\psi_j \tag{11}$$
$$\text{s.t.} \quad \beta \leq \sum_j F_{nj}\psi_j - \sum_j F_{kj}\psi_j \qquad \forall k \in [n-1]$$

For any given $\psi$, the optimal value of $\beta$ is:

$$\beta^* = \min_{k \in [n-1]} \left( \sum_j F_{nj}\psi_j - \sum_j F_{kj}\psi_j \right)$$
$$= \sum_j F_{nj}\psi_j - \max_{k \in [n-1]} \sum_j F_{kj}\psi_j.$$

Therefore, eq. (11) is equivalent to:

$$\min_{\psi \in [0,1]^m} \quad (c_n - c')\frac{\sum_j F_{nj}\psi_j}{\sum_j F_{nj}\psi_j - \max_k \sum_j F_{kj}\psi_j}. \tag{12}$$

Divide the numerator and the denominator by $\sum_j F_{nj}\psi_j$ to obtain the transformed objective:

$$\frac{\sum_j F_{nj}\psi_j}{\sum_j F_{nj}\psi_j - \max_{k \in [n-1]} \sum_j F_{kj}\psi_j} = \frac{1}{1 - \max_{k \in [n-1]} \frac{\sum_j F_{kj}\psi_j}{\sum_j F_{nj}\psi_j}}$$
$$= \frac{1}{1 - \max_{k \in [n-1]} \rho_k(\psi)}.$$

And hence eq. (12) can be written compactly as:

$$\min_{\psi \in [0,1]^m} \quad \frac{c_n - c'}{1 - \max_{k \in [n-1]} \rho_k(\psi)}. \tag{13}$$

The optimal solution for eq. (13) is the minimizer of $\max_{k \in [n-1]} \rho_k(\psi)$, which is equivalent to the minimax ratio-optimal test $\psi_\rho^*$ by Definition 2. The optimal expected pay is $\frac{c_n - c'}{1 - \rho^*}$, and the optimal contract is given by:

$$t^* = \frac{c_n - c'}{\beta^*} \psi^* = \frac{c_n - c'}{\mathrm{TP}(\psi_\rho^*) - \mathrm{FP}(\psi_\rho^*)} \psi_\rho^* = B_\rho^*(c_n - c') \cdot \psi_\rho^*,$$

where $B_\rho^*(\cdot)$ is as in Definition 3. We conclude that $t^*$ is the ratio-optimal statistical contract, as required. □

### D.4  Proof of main theorem

We are now ready to prove our main theorem:

*Proof of Theorem 1.* We prove the first half of the theorem, i.e., that the sum-optimal statistical contract is $b$-cost-robust and has the lowest budget $b/(1 - R^*)$ among all $b$-cost-robust contracts. The second half of the theorem follows by swapping Lemma 2 with Lemma 3.

We first show that the sum-optimal statistical contract is $b$-cost-robust, i.e., incentivizes the target action for every cost vector in the ambiguity set $\mathcal{C}$: Define $c^0 := (0, \ldots, 0, b)$. By Lemma 2, the min-budget contract for the setting $(F, c^0)$ is the sum-optimal statistical contract $t_R^*$, i.e., $B_R^*(c_n - c') \cdot \psi_R^*$, where $\psi_R^*$ is the minimax sum-optimal composite test for distribution sets $\{F_i\}_{i \in [n-1]}$, $\{F_n\}$. Its budget is $(b - 0)/(1 - R^*) = b/(1 - R^*)$.

In particular, $t_R^*$ incentivizes the target action and so belongs to $\mathcal{T}_{(F, c^0)}$. Observe that any increasing cost vector $\bar{c}$ with $\bar{c}_n = b$ dominates the cost vector $c^0$, and therefore by Lemma 1, contract $t_R^*$ also belongs to $\mathcal{T}_{(F, \bar{c})}$ for any such cost vector $\bar{c}$ that dominates $c^0$. Furthermore, any cost vector $c$ in the ambiguity set $\mathcal{C}$ has a corresponding cost vector $\bar{c}$ in which all costs are identical except for $c_n \leq \bar{c}_n = b$. Lowering the target action's cost from $\bar{c}_n$ to $c_n$ can only help incentivize it, thus we conclude that $t_R^* \in \mathcal{T}_{(F, c)}$, as required.

We now show optimality of the budget: Since $t_R^*$ is the min-budget contract for the setting $(F, c^0)$, and since $c^0$ is within the ambiguity region $\mathcal{C}$, it holds that any $b$-cost-robust contract $t$ must satisfy $B_t \geq b/(1 - R^*)$. As $t_R^*$ satisfies this bound exactly, it has the lowest budget among all $b$-cost-robust contracts. □

### D.5  Proof of properties of optimal cost-robust contracts

*Proof of Theorem 2.* Let $c^{c_n - a} = (c_n - a, \ldots, c_n - a, c_n)$ and $c^{c_n - b} = (c_n - b, \ldots, c_n - b, c_n)$ be two uniform-cost profiles; for brevity we refer to these as $c^{-a}, c^{-b}$. Since in contract setting $(F, c)$ it holds that $(c_n - c_i) \in [a, b]$ for all $i \in [n - 1]$, we have that $c_i^{-b} \leq c_i \leq c_i^{-a}$ for all $i$. Thus by Lemma 1 it holds that

$$\mathcal{T}_{(F, c)} \subseteq \mathcal{T}_{(F, c^{-a})}, \tag{14}$$

that is, any contract that incentivizes the target action in setting $(F, c)$ will incentivize it also in setting $(F, c^{-a})$.

Consider the min-budget contracts for settings $(F, c^{-a}), (F, c), (F, c^{-b})$. Denote their budgets by $B_{(F, c^{-a})}^*, B_{(F, c)}^*, B_{(F, c^{-b})}^*$, respectively. We deduce from Equation (14) that

$$B_{(F, c)}^* \geq B_{(F, c^{-a})}^*, \tag{15}$$

since the min-budget contract for $(F, c)$ is feasible for $(F, c^{-a})$. Now recall that Lemma 2 gives us an expression for the optimal budgets of the two uniform-cost settings. This expression depends on the risk $R^*$ of the minimax sum-optimal hypothesis test for $\{F_k\}_{k=1}^{n-1}, \{F_n\}$, which is static across the two settings. It also depends on the difference between the highest and lowest cost in each setting. Thus:

$$\frac{B_{(F, c^{-a})}^*}{B_{(F, c^{-b})}^*} = \frac{a}{b}. \tag{16}$$

Combining Equation (15) and Equation (16) we get $\frac{b}{a} B_{(F, c)}^* \geq B_{(F, c^{-b})}^*$. We conclude that the min-budget contract for setting $(F, c^{-b})$ is a $\frac{b}{a}$-min-budget contract for $(F, c)$. By Lemma 2 the

min-budget contract is the sum-optimal statistical contract $\frac{b}{1-R^*}\psi_R^*$, which by Theorem 1 is the $b$-cost-robust contract with the lowest budget, as required.

We now turn to the claim of tightness. Consider the following contract design setting:

$$F_1 = (1, 0)$$
$$F_2 = (\varepsilon, 1 - \varepsilon)$$
$$F_3 = (0, 1)$$

Where costs are increasing $c_1 < c_2 < c_3$, and $\varepsilon$ is a parameter satisfying:

$$\varepsilon < \frac{c_3 - c_2}{c_3 - c_1}. \tag{17}$$

The target distribution $F_3$ is only supported on $j = 3$, and therefore the minimax sum-optimal test for this setting is:

$$\psi^* = (0, 1).$$

As $F_1, F_3$ do not overlap, the minimax risk is given with respect to $F_2$ by the Neyman-Pearson Lemma [29]:

$$R = 1 - \|F_2 - F_3\|_{\mathrm{TV}} = 1 - \varepsilon,$$

and therefore the approximate contract given by Theorem 2 is:

$$t = \frac{c_n - c_1}{1 - R}\psi^* = \left(0, \frac{c_3 - c_1}{\varepsilon}\right).$$

As for the optimal contract, it satisfies $t^* = (0, B^*)$ because the target distribution is only supported on $j = 2$, and it has a threshold form due to [31, Lemma 4]. The optimal budget is:

$$B^* = \max\left\{c_3 - c_1, \frac{c_3 - c_2}{\varepsilon}\right\}$$

When $\varepsilon$ satisfies Equation (17), the optimal budget is $B^* = \frac{c_3 - c_2}{\varepsilon}$, and therefore:

$$\begin{aligned}
\|t\|_\infty &= \frac{c_3 - c_1}{\varepsilon} \\
&= \frac{c_3 - c_1}{c_3 - c_2} \cdot \underbrace{\frac{c_3 - c_2}{\varepsilon}}_{=B^*} \\
&= \frac{c_n - c_1}{c_n - c_{n-1}}\|t^*\|_\infty
\end{aligned}$$

as required. $\square$

**Remark 1** (Extending the proof of Theorem 2 to cost-robust min-pay contracts)**.** *By Lemma 3, the min-pay contracts for the design settings $(F, c^{-a}), (F, c^{-b})$ defined in the proof depend linearly on $a, b$, respectively, and thus their expected pay is also linear in the bounds. The inclusion argument in eq. (14) holds for min-pay contracts as well, and thus an argument analogous to eq. (16) can be constructed for the ratio of expected payments. The rest of the proof follows similarly.*

*Proof of Proposition 1.* By Theorem 1 and Lemma 2, the $b$-cost-robust contract with minimum budget is the min-budget contract for setting $(F, c^{\mathrm{const}})$ where $c^{\mathrm{const}} = (0, \ldots, 0, b)$. For this setting, plugging variables $\tilde{\lambda}, \tilde{\mu}$ into the dual in eq. (6), we get:

$$\min_{\tilde{\lambda}\in\mathbb{R}_{\geq 0}^{n-1},\tilde{\mu}\in\mathbb{R}_{\geq 0}^m} \sum_{j\in[m]} \tilde{\mu}_j$$
$$\text{s.t.} \quad \sum_{i<n}(F_{n,j} - F_{i,j})\tilde{\lambda}_i \leq \tilde{\mu}_j \qquad \forall j \in [m]$$
$$\sum_{i<n} b\tilde{\lambda}_i \geq 1$$

Define the following variable transformation:

$$\lambda = b\tilde{\lambda}; \quad \mu = b\tilde{\mu}.$$

Under this transformation, the dual LP is equivalent to:

$$\min_{\lambda \in \mathbb{R}_{\geq 0}^{n-1}, \mu \in \mathbb{R}_{\geq 0}^m} \quad \sum_{j \in [m]} \mu_j$$
$$\text{s.t.} \quad \sum_{i < n} (F_{n,j} - F_{i,j}) \lambda_i \leq \mu_j \qquad \forall j \in [m]$$
$$\sum_{i < n} \lambda_i \geq 1$$

When the contract is implementable, the optimal solution to the primal statistical LP (eq. (3)) satisfies $\beta^* > 0$, corresponding to the last constraint in the dual LP. Therefore, the last constraint of the dual is tight due to complementary slackness:

$$\sum_{i < n} \lambda_i = 1,$$

and the LP is equivalent to:

$$\min_{\lambda \in \Delta([n-1]), \mu \in \mathbb{R}_{\geq 0}^m} \quad \sum_{j \in [m]} \mu_j$$
$$\text{s.t.} \quad F_{n,j} - \sum_{i < n} \lambda_i F_{i,j} \leq \mu_j \qquad \forall j \in [m]$$

As $\mu \geq 0$ we can write:

$$\min_{\lambda \in \Delta([n-1])} \quad \sum_{j \in [m]} \left( F_{n,j} - \sum_{i < n} \lambda_i F_{i,j} \right)^+,$$

and by the definition of total variation distance (e.g by [31, Claim 4]), the optimization objective satisfies:

$$\sum_{j \in [m]} \left( F_{n,j} - \sum_{i < n} \lambda_i F_{i,j} \right)^+ = \left\| F_n - \sum_i \lambda_i F_i \right\|_{\text{TV}}$$

Applying the inverse transformation yields:

$$\sum_{j \in [m]} \tilde{\mu}_j^* = \frac{1}{b} \sum_{j \in [m]} \mu_j^*$$
$$= \frac{\min_{\lambda \in \Delta([n-1])} \left\| F_n - \sum_{i < n} \lambda_i F_i \right\|_{\text{TV}}}{b}.$$

Then, by strong LP duality $\beta^* = \sum_j \tilde{\mu}_j^*$, and the final result is obtain by applying the nonlinear variable transformation $B^* = \frac{1}{\beta^*}$. This gives the desired expression for the minimum budget $B^*$ of a $b$-cost-robust contract: $\max_{\lambda \in \Delta([n-1])} b / \left\| F_n - \sum_{i < n} \lambda_i F_i \right\|_{\text{TV}}$. $\qquad \square$

## D.6 MLR

In this section, we prove that cost-robust min-budget contracts for distributions satisfying the Monotone Likelihood Ratio (MLR) property have a threshold functional form.

*Proof of Proposition 2.* Let $(F, c)$ be a contract design setting with $c_n - c_a \leq b$, such that $F$ satisfies monotone likelihood ratio. By the Karlin-Rubin theorem, the hypothesis test for the composite hypotheses $\{F_k\}_{k=1}^{n-1}, \{F_n\}$ minimizing FP + FN is a threshold function, and therefore there exists $j_0 \in [m]$ such that $\psi^*(j) = \mathbb{1}[j \geq j_0]$. By Theorem 1, the optimal contract in this case is of the form $t^* = B\psi^*$ for some scalar $B > 0$, and therefore $t^*$ is a threshold contract. $\qquad \square$

### D.7 Two-outcome settings

**Proposition 9.** *Let $(F, c)$ be a two-outcome contract design setting ($m = 2$). A contract $t = (t_0, t_1)$ with $t_1 \geq t_0$ implements the target action if and only if the contract $t' = (0, t_1 - t_0)$ implements the target action.*

*Proof.* For any action $i \in [n-1]$, the corresponding (IC) constraint is:

$$\sum_{j \in \{0,1\}} f_{i,j} t_j - c_i \leq \sum_{j \in \{0,1\}} f_{n,j} t_j - c_n \tag{18}$$

As $f_i$, $f_n$ are probability distributions, the following holds for any $t_0$:

$$\sum_{j \in \{0,1\}} f_{i,j} t_0 = \sum_{j \in \{0,1\}} f_{n,j} t_0 \tag{19}$$

Subtracting eq. (19) from eq. (18) does not change the (IC) constraint, as both sides of eq. (19) are equal. Performing the subtraction and rearranging the terms gives:

$$\sum_{j \in \{0,1\}} f_{i,j} (t_j - t_0) - c_i \leq \sum_{j \in \{0,1\}} f_{n,j} (t_j - t_0) - c_n$$

which is equivalent to:

$$f_{i,1}(t_1 - t_0) - c_i \leq f_{n,1}(t_1 - t_0) - c_n$$

and this is the (IC) constraint for the contract $t' = (0, t_1 - t_0)$. Therefore, the contract $t$ is feasible if and only if the contract $t'$ is feasible. $\square$

**Proposition 10.** *Let $(F, c)$ be a two-outcome contract design setting ($m = 2$), and let $t = (t_0, t_1)$. Then the contract $t' = (0, t_1 - t_0)$ has weakly-better expected pay, weakly-better budget requirements, and the same variance as $t$.*

*Proof.* For the min-pay objective, we obtain from linearity of expectation:

$$\mathbb{E}_{j \in f_n}[t_j - t_0] \leq \mathbb{E}_{j \in f_n}[t_j]$$

and therefore $t'$ has weakly-better expected pay. For the min-budget objective, it holds that :

$$\max\{0, t_1 - t_0\} \leq \max\{t_0, t_1\}$$

and therefore $t'$ has weakly-better budget requirement. for the min-variance objective, adding a constant to a random variable does not affect its variance:

$$\mathrm{Var}(t) = \mathrm{Var}(t')$$

and therefore $t'$ has the same variance as $t$. $\square$

*Proof of Proposition 3.* For all $i \in [n]$, let $F_i = \mathrm{Bernoulli}(p_i)$. As $m = 2$, any contract $t$ is a two-dimensional vector. By proposition 9, proposition 10, it holds that the optimal contract is of the form $t^* = (0, t_1^*)$ for any of the three objectives. To prove that the optimal payment $t_1^*$ is the same for all objectives, observe that all objective functions are monotonically-increasing in $t_1^*$:

$$\max t^* = t_1^* \qquad \text{(Required budget)}$$
$$\mathbb{E}_{j \sim f_n}[t^*] = f_{n,1} t_1^* \qquad \text{(Expected pay)}$$
$$\mathrm{Var}(t^*) = f_{n,1}(1 - f_{n,1})(t_1^*)^2 \qquad \text{(Variance)}$$

Since all the optimization problems are of a single variable with identical (IC) constraints, their optimal solutions are all identical. $\square$

| Model size | Llama2 cost ($/1M tokens) | CodeLlama cost ($/1M tokens) |
|:---:|:---:|:---:|
| 7B | $0.182 | $0.183 |
| 13B | $0.24 | $0.24 |
| 70B | $0.64 | *$0.64* |

Table 3: Estimated energy costs for the Llama2 and CodeLlama model families, according to the methodology described in Appendix E.1.

### D.8 Hardness of all-or-nothing cost-robust contracts

While cost-robust contracts can be computed in polynomial time by solving the corresponding convex programs (see Appendix D.1), restricting the functional form of the solution to have only two levels of payment entails computational hardness:

**Definition 6** (All-or-nothing contract; [31]). *A contract $t$ has an* all-or-nothing *functional form if there exists $B > 0$ such that $t_j \in \{0, B\}$ for all $j \in [m]$.*

In [31, Theorem 3], it is shown that computing a min-budget all-or-nothing contract is NP-hard. We show that the same reduction is applicable for min-budget cost-robust contracts:

**Proposition 11** (Hardness). *Computing a min-budget all-or-nothing contract is NP-hard.*

*Proof.* By reduction from 3SAT. Given a 3-CNF formula, we show that there exists a cost-robust contract design problem such that the required budget of an all-or-nothing cost-robust min-budget contract is below a certain threshold if and only if the formula is satisfyiable.

First, given a 3-CNF formula, apply the polynomial-time reduction described in [31, Appendix B.5.3] to construct a min-budget contract design problem $(F, c)$. Denote the target action of the design problem by $n \in [n]$. By the construction described in [31, Equation (27)], it holds that $c_n = 1$ and $c_i = 0$ for all $i < n$. Thus, the contract design problem tightly satisfies the cost difference upper bound $c_n - c_i \leq 1$. Moreover, since all alternative actions have the same cost $c_i = 0$, the cost vector $c$ it is also the worst-case cost vector in the cost uncertainty set induced by the bound $b$ (by Lemma 1). Therefore, $(F, c)$ is also a cost-robust contract design setting for the bound $c_n - c_i \leq 1$.

By the proof of [31, Theorem 3], there exists a threshold $B_0$ such that the required budget of an all-or-nothing min-budget contract in the design setting $(F, c)$ is below a certain threshold if and only if the 3-CNF formula is satisfiable. $\square$

## E Experiments/Empirical Evaluation

### E.1 Inference Costs

To calculate the costs for each model, we use energy data from the Hugging Face LLM Performance Leaderboard[5]. The energy efficiency for each model is expressed in the leaderboard in units of output tokens per kWH. To convert to actionable costs we assume a rate of .105 $/kWH, aligning with conservative energy costs in the United States and giving us order-of-magnitude approximations of the actual inference costs. The inference costs are presented in units of $/1M tokens. The leaderboard data was taken from the experiments on the A100 GPUs. For each model, we took the GPTQ-4bit+exllama-v2 quantization benchmark. Table 3 shows the energy costs on the leaderboard for Llama2 and CodeLlama. We note that energy data was missing for CodeLlama-70B, so we extrapolated from Llama2-70B-chat.

**Model verbosity.** Calculation of the per-token inference costs are not complete without an analysis of the response length produced by the various models. Table 4 shows the average verbosity (output length) of the 3 Llama models on the single-turn prompts in the MT-bench evaluation set. Since the values are of the same order of magnitude, we simplify and assume that the choice of model does not influence the verbosity, and therefore we do not include this in our cost calculations.

---

[5]https://huggingface.co/spaces/optimum/llm-perf-leaderboard.

| Model | Verbosity |
|---|---|
| `Llama-2-7B-chat` | 1625 |
| `Llama-2-13B-chat` | 1573 |
| `Llama-2-70B-chat` | 1695 |

Table 4: Model verbosities (average output length) of the models in consideration. Since the values are of the same order of magnitude, we assume for simplification that the choice of model does not significantly affect verbosity (see Appendix E.1).

| | Cost-aware | | | Cost-robust | | |
|---|---|---|---|---|---|---|
| | $\mathbb{E}[t]$ | $\max t_j$ | $\mathrm{stdev}(t)$ | $\mathbb{E}[t]$ | $\max t_j$ | $\mathrm{stdev}(t)$ |
| Min-Pay | **0.86** | 73.4 | 7.63 | 0.92 (+6.9%) | 73.4 | 8.16 |
| Min-Budget | 2.48 | **3.59** | 1.60 | 2.78 | 3.91 (+8.7%) | 1.70 |
| Min-Variance | 3.52 | 6.31 | **1.45** | 3.84 | 6.58 | 1.53 (+6.5%) |

Table 5: Cost-aware vs Cost-robust contracts in the non-monotone setting. The numbers in bold denote the optimal values achieved for the 3 objectives. The percentages denote the relative increase from the optimal that the cost-robustness sacrifices in each setting.

**Current market pricing schemes**   As described in Section 1, current market pricing schemes for LLM generation involve *pay-per-token* rates for which the user pays regardless of the response quality. For open-source models such as `Llama2`, there exist API services to run model inference, such as AWS and Microsoft Azure. In other scenarios, some pricing schemes behave as threshold contracts: an unsatisfied user may request from the API to regenerate a response free of charge, and hence will only pay if the response quality is above some threshold. For this reason threshold contracts can offer a "satisfaction guarantee" while retaining a simple form.

### E.2   Multi-outcome contracts: Further Analysis

**Non-Monotone Contracts**   Table 5 shows the statistics of the various contract objectives in contracts when optimized without a monotonicity constraint, and displays how they match up to each other in cost-aware and cost-robust settings. We observe that the min-pay contract minimizes expected pay at the expense of high budget and variance. The min-budget contract, on the other hand, is not the worst in any of the objectives. Additionally, the cost-robust setting sacrifices only a marginal increase in objective values: a 6.9% increase in the Min-pay objective, an 8.7% increase if optimizing for budget, and 6.5% increase when optimizing for minimum variance.

**Price of monotonicity**   It is of interest to analyze the relative difference in resulting contracts that occurs due to removing the monotonicity constraint. Table 6 shows the discrepancy in contract objectives for cost-robust contracts. We can observe that while the monotone contracts as a whole are simpler, more intuitive, and closely resemble threshold contracts, it is not without cost as they suffer a sizeable increase in contract objectives, most notably an increase of 388% when trying to minimize expected pay.

| | Non-Monotone | | | Monotone | | |
|---|---|---|---|---|---|---|
| | $\mathbb{E}[t]$ | $\max t_j$ | $\mathrm{stdev}(t)$ | $\mathbb{E}[t]$ | $\max t_j$ | $\mathrm{stdev}(t)$ |
| Min-Pay | **0.92** | 73.4 | 8.16 | 4.82 | 12.23 | 5.98 |
| Min-Budget | 2.78 | **3.91** | 1.70 | 5.48 | 6.63 | 1.83 |
| Min-Variance | 3.84 | 6.58 | **1.53** | 5.48 | 6.63 | 1.83 |

Table 6: Cost-robust contracts, monotone vs. non-monotone setting. The numbers in bold denote the optimal values achieved for the 3 objectives.

### E.3 Implementation details

**Code.** We implement our code in Python. Our code relies on `cvxpy` [12, 2] and `Clarabel` [16] for solving linear and quadratic programs.
Code is available at: `https://github.com/edensaig/llm-contracts`.

**Hardware.** All experiments were run on a single Macbook Pro laptop, with 16GB of RAM, and M2 processor, and with no GPU support. Overall computation time is approximately one minute.

**Implementation of cost-robustness.** To implement cost-robustness of a contract setting with costs $(c_1, c_2, \ldots, c_n)$, we assume knowledge of only the *range* of costs, and calculate the contract using costs $(0, 0, \ldots, c_n - c_1)$. This modeling assumption provides us with the flexibility of solving contracts in settings without full-information while maintaining the approximation guarantee set forth in Theorem 2.

#### E.3.1 Contract design solvers

To compute optimal contracts, we implemented the following solvers:

- **Convex programming solvers:** Given outcome distributions $\{f_i\}$ and costs $\{c_i\}$, we solve the MIN-PAY LP (eq. (4)), the MIN-BUDGET LP (eq. (2)), and the MIN-VARIANCE QP (eq. (9)) using `cvxpy`. All optimization programs enforce incentive compatibility (IC) constraints for the target action $n$ against all other actions $i \in [1, n-1]$. We note that the `Clarabel` solver supports both linear and quadratic programs.

- **Threshold contract solver:** To find threshold contracts for problems with low-dimensional outcome and action spaces (such as with MT-bench), we implement a brute-force solver which performs full enumerations of all possible thresholds, as proposed by [31]. We refer to the `Full Enumeration Solver` in [31, Appendix C.2.1] for further implementation details.

