# OpenReview forum: "Incentivizing Quality Text Generation via Statistical Contracts"
_NeurIPS.cc/2024/Conference — NeurIPS 2024 poster_

### Official Review · Reviewer_Bg3E · 2024-07-12

**Soundness:** 4
**Presentation:** 4
**Contribution:** 4
**Rating:** 7
**Confidence:** 3

**Summary:**

The authors formulate a theoretical setup for a LLM text generation service to incentivize the service to output high quality text the consumer. The authors formulate this setup as having the service having a set of models that has quality (as rated by a evaluator on the end of the consumer) that increases with the cost of running the LLM. The goal is to derive a framework for paying the LLM service based on that quality of the text generated that incentivizes the LLM service to always use its best model. The authors go about this by formulating the definition of a contract in this setting, and defining various metric (max payment, avg. payment, etc.) that consumer aims optimize. They show that the set of contracts that will incentivize the LLM service to output text with the best model can be derived from the set of optimal hypothesis tests that distinguish which model is being used from the evaluator outputs. They derive how the optimal contract can be formulated from these hypothesis tests, when only bounds are known on the costs of running each model for the LLM service.

**Strengths:**

The theoretical setup and monotone assumption of model performance, cost, etc. is quite reasonable and tackles and interesting and relevant problem with LLM queries. The results are simple and intuitive, and connect nicely with previous work on contract theory and hypothesis testing.

**Weaknesses:**

The main issue is the theoretical setup does require an assumption that the bounds on cost are known, which seems somewhat impractical. It might be useful to explicitly comment on how the different metrics degrade with increased looseness of the cost bounds (linearly, it seems like), since one can always pick extremely conservative cost bounds.

Minor issue:

- In Definition 3, it would be helpful to illustrate why $B_R^*$ and $B^*_\rho$ are defined as they are, correctly. Further, the definition with $b \geq c_n$ is a bit strange, since the definition of minmax hypothesis test does not involve worst case over cost vectors, so it doesn't seem correct to use $b$ to derive the minimax contract (and instead it should remain a function of $c_n - c_1$) --- maybe dropping that $b \geq c_n$ case would be more accurate, since it is used in the definition of cost-robust later.

**Questions:**

Should the principal know the outcome distribution for each generator? This seems slightly unrealistic since consumers only ever have black box access to the API, and never know precisely which model they're responses are from. They do know that previous agent actions could only be mixed over the old (worse) models though.

**Limitations:**

I think the limitations are sufficiently addressed.

---

> ### Author Rebuttal · Authors · 2024-08-07
>
> Thank you for the insightful and encouraging review! We address your questions and remarks below:
>
> > **The main issue is the theoretical setup does require an assumption that the bounds on cost are known, which seems somewhat impractical.**
>
> This is a very good point. We note that virtually all robust optimality results in the mechanism design literature assume the principal knows something about the setting (e.g., a subset of available actions in [1], or the first moments of the distributions in [2]), and we believe that supporting uncertainty regarding cost is certainly better than assuming full knowledge as in the standard contract setting. We hope that the characterization of cost-robust contracts can serve as a building block towards the development of contracts which are robust in additional ways.
>
> > **It might be useful to explicitly comment on how the different metrics degrade with increased looseness of the cost bounds (linearly, it seems like), since one can always pick extremely conservative cost bounds.**
>
> We will gladly add the analysis you suggest of degradation with looseness of the cost bounds (it is indeed linear).
>
> > **Minor issue in Definition 3: it would be helpful to illustrate why $B_R$ and $B_\\rho$ are defined as they are...**
>
> We will also revise Definition 3 - thank you for these suggestions!
>
> > **Should the principal know the outcome distribution for each generator? This seems slightly unrealistic since consumers only ever have black box access to the API, and never know precisely which model they're responses are from. They do know that previous agent actions could only be mixed over the old (worse) models though.**
>
> This is an excellent question, which suggests a very interesting direction for future research. In our model, the principal is indeed assumed to know the score distribution of the generators, and indeed can never know (without further assumptions) whether the agent “cheated” in the past to influence this knowledge.
>
> It seems that to gain from such cheating, responses should come from a weaker generator, to appear as if easy tasks require more effort from the agent than they actually do, and thus justify more compensation from the principal. In mechanism design, this is somewhat similar to systematically underbidding in auctions, so that the auctioneer will charge less in a future auction, never finding out the true value distribution of the bidder. From the game-theoretic perspective, formulating and studying a model with such non-myopic agent behavior is a very interesting question for future research. From the statistical learning perspective, it will be very interesting to identify distributional assumptions which are reasonable in practice, and also provide meaningful guarantees for learning from samples.
>
> --
>
> Again, we would like to thank you for the insightful feedback! If any additional questions or thoughts arise, please do not hesitate to let us know.
>
> References:
> * [1] Carroll, Robustness and Linear Contracts, AER 2015.
> * [2] Dutting et al., Simple vs Optimal Contracts, EC 2019.

---

### Official Review · Reviewer_4RXi · 2024-07-14

**Soundness:** 3
**Presentation:** 3
**Contribution:** 3
**Rating:** 5
**Confidence:** 3

**Summary:**

This paper addresses the issue of moral hazard in pay-per-token pricing for large language model (LLM) services. Firms may use cheaper, lower-quality models to cut costs, compromising text quality. Moreover, the firms's costs may be unknown to the clients. To counter this, the authors propose a pay-for-performance framework using cost-robust contracts that incentivize high-quality text generation under the uncertainty about the firms's costs. These contracts are designed based on and have a one-to-one correspondence to optimal composite hypothesis tests. Approximation guarantees are provided with respect to the optimal contracts. Empirical evaluations show that these contracts provide robust incentives with small cost increases.

**Strengths:**

The results of characterizing the forms of optimal cost-robust contracts using hypothesis testing, as well as the approximation guarantees, are interesting and have valuable contributions.

**Weaknesses:**

1. The model's complexity is unnecessary. The problem could actually be studied in the most basic contract setting.
2. The authors do not discuss the computational complexity of finding the optimal cost-robust contract.
3. The authors do not discuss the cases where the action with the highest cost may not be the best action to incentivize.

**Questions:**

1. what is the computational complexity of finding the optimal cost-robust contract that incentivizes $c_n$ (equivalently, the complexity of finding the optimal test)?
2. Can the characterization results and computational efficiencies be directly applied to the cases where actions with lower costs may be the best action to incentivize?

**Limitations:**

Yes.

---

> ### Author Rebuttal · Authors · 2024-08-07
>
> Thank you for the insightful review, and for the excellent questions!
>
> > **What is the computational complexity of finding the optimal cost-robust contract that incentivizes (equivalently, the complexity of finding the optimal test)?**
>
> This is a very good question. Optimal cost-robust contracts (and equivalently, optimal hypothesis tests) can be found by solving linear programs, and can therefore be computed in $\\mathrm{poly}(n,m)$ time, where $n$ is the size of the action space (number of possible text generators), and $m$ is the size of the outcome space (number of possible evaluation outcomes). This is a corollary of Theorem 1 and Lemmas 2-3:  By Theorem 1, the optimal cost-robust contract is equivalent to an optimal statistical contract, and by the lemmas this in turn is equivalent to a min-budget or min-pay contract in an appropriate uniform-cost setting. Equations (1) and (3) encode min-budget and min-pay contract LPs, respectively.
>
> Further expanding on this point, it is also worth noting that while all optimal contracts presented in this paper can be computed in polynomial time, a common theme in the contract design literature suggests that complexity may change when restricting optimization to contracts with a "simple" functional form (for example, when restricting optimal contracts to have only two levels of payment). Following your remark, we looked into the hardness results of Saig et al. [1] for the full-information min-budget setting, and found that their result is also applicable to cost-robust contracts when their functional form is restricted to have only two levels of payment (Theorem 3 in Saig et al. [1] shows that computing a min-budget contract with two levels of payment is NP-hard in the full-information setting). We are thankful for this remark, and will include this additional insight in the paper.
>
> > **Can the characterization results and computational efficiencies be directly applied to the cases where actions with lower costs may be the best action to incentivize?**
>
> This is an excellent question. Our model indeed relies on the assumption that the output of costlier LLMs has higher quality in general. While it doesn’t capture scenarios such as LLM fine-tuning, we believe that this is a reasonable starting point.
>
> A generalization of this approach currently appears in footnote 4 in the paper. Expanding upon that point, we note that our contract design scheme can be naturally generalized to scenarios where the agent seeks to incentivize any LLM above a certain quality threshold, rather than the single most costly LLM. Taking OpenAI’s products as an example, this corresponds to cases where the agent would like to incentivize text generation using GPT-3 and above, rather than GPT-3 strictly. This generalization still maintains the correspondence between cost-robust contracts and hypothesis tests, and the contract is computed by iteratively solving cost-robust design problems with a single target action. We will update the paper to include a detailed discussion of these aspects.
>
> Beyond this extension, our main analysis technique is not directly applicable, as it requires a separation between the interval covering the costs of target actions and the interval covering the costs of alternative actions (see Lemmas 2-3). In this regard, extending cost-robustness to problems with cost interval overlap seems to be a very interesting direction for future research. We will add to the paper a discussion of our theoretical techniques and their potential applicability in more general scenarios, as well as counterexamples that illustrate limitations.
>
> > **The model's complexity is unnecessary. The problem could actually be studied in the most basic contract setting.**
>
> We are not sure what “most basic'' means here and apologize in advance if we misunderstood, but if you mean a setting in which the costs are known, we note that many factors affecting cost (such as model architecture, batching policy, and exact energy costs) remain undisclosed in practical commercial applications, motivating the development of cost-robust contracts. If this wasn’t the intention, please let us know. There is always a trade-off between simplicity and expressiveness when defining models, and having a concrete reference for a basic model will help us make a more precise comparison and clarify the relations.
>
> –
>
> Please let us know if our response has addressed your questions regarding computational complexity, multiple target actions, and the conciseness of the model. If you have any further questions or thoughts, we are more than happy to clarify and discuss!
>
> Reference:
> * [1] Saig et al., Delegated Classification, NeurIPS 2023.

---

> > ### Comment · Reviewer_4RXi · 2024-08-11
> >
> > Thanks for the responses. The computational efficiency is indeed helpful. Here are my further responses and questions:
> >
> > 1. Just to clarify. Do you make any assumption on the outcome distributions related to actions with different costs? Or do you just refer the action with the highest quality to the action with the highest cost? If this is the case, why do your results only rely on incentivizing actions with the highest costs without any assumptions made on the distributions? Could you provide more intuitions and explanations for it?
> > 2. By saying basic contract problem, I mean that the problem can be studied in the basic contract setting without the modeling of LLM and Text Generation. The problem is actually a robust contract design setting where the action costs are uncertain, and LLM is just an application. Using the language of contract design with moral hazard would significantly simplify the notations.
> >
> > One minor:
> > 1. It seems that the references are not updated. the reference of Saig et al'23 is still the Arxiv one.

---

> ### Author Response · Authors · 2024-08-11
>
> Thank you for the thoughtful response! We address your remarks below:
>
> > **Do you make any assumption on the outcome distributions related to actions with different costs? Or do you just refer the action with the highest quality to the action with the highest cost? If this is the case, why do your results only rely on incentivizing actions with the highest costs without any assumptions made on the distributions? Could you provide more intuitions and explanations for it?**
>
> This is a great question. Our main theoretical results only assume that the target action is implementable - i.e., that there exists some contract incentivizing it (see, e.g., Appendix B.2). Intuitively, implementability is equivalent to the assumption that the observed quality of the target LLM is different (in distribution) from the observed quality of smaller, cheaper models [1]. One intuition for the fact that no further distributional assumptions are required is the equivalence to optimal composite hypothesis testing (by Theorem 1), which doesn't require further assumptions as well.
>
> We also note that the implementability assumption can be verified in polynomial time given outcome distributions, by checking the feasibility of the corresponding linear programs (i.e. equations (1,3)). Additionally, the assumption was verified to hold in the empirical datasets we analyze. In scenarios where the highest-cost action cannot be implemented by any contract (i.e., when the observed quality of the costly LLM is identical to the quality of cheaper ones), this action can be ignored, and the next highest-cost action effectively becomes the highest-cost one and can be targeted instead. We will further emphasize these points in the paper.
>
> Further extending on this remark, we also note that another common theme in the literature is providing stronger guarantees on the resulting contracts by making stronger assumptions about the structure of outcome distributions (see, e.g., [1,2]). Connecting to this theme, in Proposition 2 we show that the MLR structural assumption (Monotone Likelihood Ratio, Def. 5) implies a threshold functional form for the optimal cost-robust contracts. However, from the empirical perspective, it is also worth noting that the outcome distributions we observe in our empirical study don’t seem to satisfy the theoretical assumptions currently available in the literature (for example, see Figure 2 middle right, which shows the non-trivial outcome distributions of the MT-Bench dataset). In this context, we hope that our empirical observations will motivate future theoretical research with refined structural assumptions. We will emphasize this point in the paper as well.
>
> > **By saying basic contract problem, I mean that the problem can be studied in the basic contract setting without the modeling of LLM and Text Generation. The problem is actually a robust contract design setting where the action costs are uncertain, and LLM is just an application. Using the language of contract design with moral hazard would significantly simplify the notations.**
>
> Thanks for the clarification, this is also a very good point. The cost-robustness model and our theoretical results indeed apply more broadly, and extend beyond the context of LLMs and text generation - We view it as one of the paper’s main strengths. Through the use of "application specific" language (i.e. focusing on LLMs), our hope is to promote discussion between different scientific communities, which will eventually increase the applicability of contracts in this setting. The LLM market is nascent and evolving - for example, more players are joining, and different pricing (contract) schemes are emerging. The current naïve pricing schemes, which do not yet tie payments to performance, are likely to be reshaped and improved. Experience from the sponsored search market shows that economic theory informs better pricing [3], and we expect that through contractual payments this will be the case for LLM pricing too. In any case, we will also update the paper to further clarify the significance of our results to general contract design theory.
>
> > **One minor: It seems that the references are not updated. the reference of Saig et al'23 is still the Arxiv one.**
>
> Thank you for this remark! We will gladly fix this.
>
> –
>
> Please let us know if our response has addressed your questions regarding distributional assumptions, and the relation to general contract design theory. Also, if the discussion so far increases the favorability of your assessment, we would greatly appreciate it if you would consider increasing your score. In any case, your remarks are very helpful, and we are more than happy to discuss any additional questions or thoughts that come up.
>
> References:
> * [1] Dutting et al., Simple vs Optimal Contracts, EC 2019.
> * [2] Saig et al., Delegated Classification, NeurIPS 2023.
> * [3] Ostrovsky and Schwarz, Reserve Prices in Internet Advertising Auctions: A Field Experiment, JPE 2023.

---

### Official Review · Reviewer_RQsK · 2024-07-14

**Soundness:** 3
**Presentation:** 3
**Contribution:** 1
**Rating:** 5
**Confidence:** 3

**Summary:**

* The paper concerns the problem of incentivizing LLMs to use the most costly model, which is assumed to be the model with the best performance. Without proper incentive, the LLM company has the incentive to charge customers for the highest payment, but deliver the service using a lower-cost model, because the performance of the model is usually not verified. Therefore, the paper proposes to use contract design to solve this problem. In particular, an automatic detector first gives an integer score for the performance of an LLM. Then, the task is to design a payment for the company for each of the integer scores. The goal is to minimize the total payment conditioned on incentivizing the best-performed model.
* The main contribution is the discussion of the cost-robust contract, meaning how to design the optimal contract while the costs of LLMs are unknown. Empirical evaluations have shown how to use the theory in practical settings given LLM performance data.

**Strengths:**

* (I’m not an expert in contract design.) I appreciate the theoretical contributions of the paper. To me, section 4 has several interesting insights into connecting cost-robust contracts with hypothesis tests. As claimed, this is the first paper considering cost-robust contract design. However, the real contribution should be better evaluated by experts.
* In general, incentive issues of LLM uses have been very critical and challenging. I also like the connection between contract design and the production of LLMs.

**Weaknesses:**

Although I believe contract design can speak with the production of LLMs, I’m not fully convinced that the proposed model is a good idea to solve the considered problem.

* In practice, each company pricing its own AIs, so who should be the principal? In other words, the paper assumes there is a trust-worthy third party who can run the quality-detector and commits to a contract with the LLM companies. I’m not sure this is feasible in practice. I hope the authors can explain more carefully the application scenarios of their theory.
* Furthermore, there is no evidence in the paper (and, I guess, on the Internet) that can prove LLM companies are cheating about their service quality. I also don’t think this is very likely because LLM companies have other incentives to provide high-quality services, e.g. their reputation. So, how do we know we are not solving a problem that does not exist?
* Even though we go with the assumption that there is a contract that the company (agent) agrees on, I doubt cost-robustness is the first-order concern. The cost data is usually publicly obtainable from energy reports, as the authors did in their experiments. Even though this data is not public, the energy cost is usually easy to estimate. Therefore, I don’t think incentivizing LLM production is a suitable application for cost-robust contracts.

**Questions:**

See weakness

**Limitations:**

Limitations are reasonably stated.

---

> ### Author Rebuttal · Authors · 2024-08-07
>
> Thanks for the insightful review! We address the points below:
>
> > **In practice, each company pricing its own AIs, so who should be the principal? In other words, the paper assumes there is a trust-worthy third party who can run the quality-detector and commits to a contract with the LLM companies.**
>
> Thank you for this question. One application our solution applies to is cases in which the principal (a consumer, or a group) has some bargaining power. While this doesn’t cover all possible scenarios, in many cases the principal is in fact in a strong position: Large organizations typically have bargaining power as they license software in bulk (through "enterprise licensing"). Additionally, our results also suggest that users with common interests are likely to benefit from negotiating together, or purchasing LLM services through agencies with negotiation power as in the online ad market. We will update the paper to emphasize these aspects.
>
> Regarding the need for third-party evaluation, it is first important to note that evaluation can be performed locally by the principal in a verifiable way, and thus there is no need for a third-party. This applies naturally in important use-cases such as LLM-assisted code generation. For evaluation settings which are more intricate, following your remark we have extended our framework to support contracts with principal-side sampling, where the principal only evaluates a fixed proportion of results that are chosen uniformly at random (e.g. 1% of responses at random). This results in a contract which requires more budget but demands less evaluation resources. Allowing this trade-off can eliminate the need for a trusted third-party in additional key use-cases, such as evaluation based on an LLM-as-a-judge that runs locally. We will add a full formal description to the paper, together with supporting proofs.
>
> Lastly, we also note that evaluator integrity is a common assumption in the contract design literature, and any economic/cryptographic solution to the evaluator integrity problem can be applied in our case as well.
>
> > **There is no evidence in the paper (and, I guess, on the Internet) that can prove LLM companies are cheating about their service quality. I also don’t think this is very likely because LLM companies have other incentives to provide high-quality services, e.g. their reputation.**
>
> While strategic misreporting in LLMs has not yet been officially acknowledged, there is ample evidence for strategic misreporting in similar industries with more mature markets, and related evidence suggesting that strategic behavior is also plausible in the context of LLMs:
>
> * In related industries, there are certainly examples of trust violations from companies who provide black box services - an extreme example is Theranos, and another example is throttling by internet service providers, which purposefully and intermittently slow down service.
> * For LLMs, there has already been a documented period of alleged low-quality service from ChatGPT around January 2024, dubbed “ChatGPT gone lazy”, which OpenAI acknowledged but never fully explained (see The Guardian, “What is going on with ChatGPT?”, Jan 12, 2024). With the current pricing schemes, it is not clear what will prevent such periods from recurring in the future.
>
> While indirect systems such as reputation or the litigation system allow for some level of quality assurance, contracts are increasingly being applied in real-world applications - such as pay-for-performance healthcare, or revenue sharing for internet content creators. For reputation, one of the issues is that quality is non-trivial to assess, and quality measures are noisy. Considering contracts in conjunction with complementary systems like reputation appears very promising for future work, and we are very grateful for this insightful remark.
>
> > **The cost data is usually publicly obtainable from energy reports, as the authors did in their experiments. Even though this data is not public, the energy cost is usually easy to estimate.**
>
> To our knowledge, fine-grained cost analysis is only publicly available for open-source models. Our experiments use data from the open-source Llama-2 models with publicly-disclosed computational setups. In contrast, the full architecture and computational setup of the commercial models common today are not publicly disclosed (for example, GPT-4, Gemini, and Claude are all closed-source). For closed models, many factors affect the cost (architecture, batching policy, GPU type, exact energy prices, etc.), and many of them remain undisclosed - Motivating cost-robustness.
>
> Moreover, there is no doubt that simplicity and explainability are important practical concerns in pricing, and our contracts based on hypothesis tests achieve exactly these properties, while also having a deep theoretical justification as optimal cost-robust contracts. This mirrors seminal results in contract theory that justify the use of simple linear contracts due to their robust optimality - whether to the full action set [1] or to the distributional details [2].
>
> –
>
> We would like to thank you again for the insightful questions and remarks! We hope this discussion clarifies the plausibility of moral hazard problems in LLMs, the applicability of contract design to them, and the significance of the contribution to the foundations of contract design theory.
>
> Please do not hesitate to follow up if you feel some questions remain unanswered.
>
> References:
> * [1] Carroll, Robustness and Linear Contracts, AER 2015.
> * [2] Dutting et al., Simple vs Optimal Contracts, EC 2019.

---

> > ### Comment · Reviewer_RQsK · 2024-08-12
> > **post rebuttal**
> >
> > I appreciate the author's rebuttal. Although I'm still not fully convinced by the rebuttal of the second point which concerns the applicability of the theory, I can see some value in the model and theoretical analysis. I'm raising my score to 5.

---

> > > ### Author Response · Authors · 2024-08-13
> > >
> > > Thank you so much for the improved assessment! We appreciate your feedback, and will certainly expand on this discussion in the paper to further facilitate the applicability of our model. We are also more than happy to continue the discussion if additional questions arise.

---

### Decision · Program_Chairs · 2024-09-25

**Decision:**

Accept (poster)

**Comment:**

The paper takes a contract design approach for LLMs, proposing a pay-for-performance, contract-based framework for incentivizing quality.

The review team (including the AC) agrees that the paper makes a solid theoretical contribution to the literature, by offering an interesting statistical characterization of optimal cost-robust contracts by connecting them to statistical hypothesis tests. This contribution is perhaps of independent interest (thus applicable beyond the LLMs setting).

On the other hand, the review team pointed out that there is little to no evidence on the misalignment of incentives in quality text generation. More generally, the application of cost-robust contract design to LLMs might seem somewhat superficial / not particularly suitable. Overall, the practical motivation of the problem and its applications can be significantly improved.

Despite these concerns, the review team weighted more positively the theoretical and conceptual contribution of the paper.